# A ZFYVE21-Rubicon-RNF34 signaling complex promotes endosome-associated inflammasome activity in endothelial cells

Xue Li[1,2,3,9], Quan Jiang [1,2,9], Guiyu Song[1,2,4,9] ✉, Mahsa Nouri Barkestani[1,2], Qianxun Wang [1,2], Shaoxun Wang[1,2], Matthew Fan [5], Caodi Fang[1,2], Bo Jiang[2,6,7], Justin Johnson[8], Arnar Geirsson[6], George Tellides[6], Jordan S. Pober [8] & Dan Jane-wit [1,2,8] ✉

Internalization of complement membrane attack complexes (MACs) assembles NLRP3 inflammasomes in endothelial cells (EC) and promotes IL-β-mediated tissue inflammation. Informed by proteomics analyses of FACS-sorted inflammasomes, we identify a protein complex modulating inflammasome activity on endosomes. ZFVYE21, a Rab5 effector, partners with Rubicon and RNF34, forming a "ZRR" complex that is stabilized in a Rab5- and ZFYVE21-dependent manner on early endosomes. There, Rubicon competitively disrupts inhibitory associations between caspase-1 and its pseudosubstrate, Flightless I (FliI), while RNF34 ubiquitinylates and degradatively removes FliI from the signaling endosome. The concerted actions of the ZRR complex increase pools of endosome-associated caspase-1 available for activation. The ZRR complex is assembled in human tissues, its associated signaling responses occur in three mouse models in vivo, and the ZRR complex promotes inflammation in a skin model of chronic rejection. The ZRR signaling complex reflects a potential therapeutic target for attenuating inflammasome-mediated tissue injury.

Inflammasomes are megadalton multimers that regulate tissue inflammation in response to a broad range of cellular stressors. Following terminal assembly, inflammasome complexes promote maturation of IL-1 family cytokines including IL-1β and IL-18. While canonically studied in macrophages, inflammasomes are assembled across a broad range of immune- and parenchymal-derived cell types relevant to auto- and alloimmune disease including endothelial cells (ECs)[1–3], synovial fibroblasts[4], epithelial cells[5], and T cells[6]. Inflammasomes and their derived products mediate systemic inflammation and are well-defined drug targets for ameliorating disease-related tissue injury[7].

Complement are ancient immune proteins involved in sensing foreign and/or damaged surfaces. In auto- and allo-immune vasculitides, complement proteins become activated principally on arterial ECs. Following activation on ECs, terminal complement proteins self-assemble into membrane attack complexes (MACs), large heterodimers which intercalate into EC surfaces as transmembrane pores. MAC deposition on ECs imparts vasculopathic properties to these same cells, promoting vasculitis and vaso-occlusive lesions in vivo. Complement activity involving MAC assembly on ECs is prognostic for, diagnostic for, and a therapeutic target for rejection of tissue allografts.

[1]VA Connecticut Healthcare System, West Haven, CT, USA. [2]Department of Cardiovascular Medicine, Yale University School of Medicine, New Haven, CT, USA. [3]Department of Nephrology, Shengjing Hospital of China Medical University, Shenyang, China. [4]Department of Obstetrics and Gynecology, Shengjing Hospital of China Medical University, Shenyang, China. [5]Yale College, Yale University, New Haven, CT, USA. [6]Dept of Surgery, Yale University School of Medicine, New Haven, CT, USA. [7]Dept of Vascular Surgery, The First Hospital of China Medical University, Shenyang, China. [8]Dept of Immunobiology, Yale University School of Medicine, New Haven, CT, USA. [9]These authors contributed equally: Xue Li, Quan Jiang, Guiyu Song. ✉e-mail: guiyu.song@yale.edu; dan.jane-wit@yale.edu

Although MACs may lyse microbes or red blood cells, nucleated cells like ECs resist cytolysis by autologous MACs via various mechanisms including both shedding of membrane vesicles and by internalization[8]. We used 'high' panel reactive antibody (PRA) sera obtained from renal transplant candidates to model the non-cytolytic properties of MACs on human ECs[2,3,9–12]. Due to risks of MAC assembly on donor ECs, transplant candidates routinely undergo PRA testing where patient sera is overlaid on a broad panel of Luminex beads, each of which is coated with a singular HLA specificity. The number of beads showing reactivity is reported as a percentage reflecting a correlative risk for prospective development of alloantibody (alloAb)-induced MAC formation on ECs upon transplantation. Due to multiple child-births, blood transfusions, and/or prior transplants, certain patients contain high titers of non-self binding alloAbs showing >80% bead binding. These patients with 'high' PRA sera are at-risk for developing MAC formation on ECs and subsequent tissue rejection. Our lab has used 'high' PRA sera, termed PRA, as an investigational tool to elicit alloAb-induced, non-cytolytic assembly of MACs on human ECs in vitro and in vivo[10].

During the course of our studies which we briefly describe below for context, we discovered that a protein called ZFYVE21 elicited inflammasome activity in association with Rab5 endosomes. PRA induces assembly of MACs at the cell surface, and MACs undergo clathrin-mediated endocytosis and transfer to Rab5+ vesicles, forming MAC+Rab5+ signaling endosomes[11]. In a process requiring NF-κB inducing kinase (NIK), inflammasome proteins including NLRP3, ASC, and caspase-1 are recruited to signaling endosomes and become juxtaposed to catalyze their oligomerization. These processes result in Rab5-associated caspase-1 activity[2]. ZFYVE21 mediated the above by enriching signaling endosomes in PI(3,4,5)P3 to allow Akt-dependent recruitment of NIK[1,9].

ZFYVE21 is a conserved protein containing an N-terminal FYVE domain that was previously found to localize to early endosomes to mediate cell motility[6]. While promoting Rab5-associated inflammasome activity by recruiting NIK[1,9], we hypothesized that ZFYVE21 could additionally modulate inflammasome activity by regulating oligomerization of inflammasome components. The strength of canonical inflammasome activity is dictated by the ability of its component protease, caspase-1, to oligomerize[13]. Homo-oligomerization of procaspase-1 results in proximity-dependent auto-catalysis and formation of the enzymatically active protease, cleaved caspase-1, whose generation is regulated by prion-like seed activity[14], post-translational modifications[15–17] and, in humans, binding to pseudosubstrates including CARD-Only Proteins (COPs)[13] as well as Flightless I (FliI)[13,15,18]. To explore the ability of ZFYVE21 to regulate oligomerization of inflammasome proteins, we formulated a bioinformatics strategy and identified Rubicon and RNF34 as proteins showing concurrent binding to Rab5 endosomes and to inflammasome proteins. Rubicon and RNF34 associated with ZFYVE21, forming a ZFYVE21-Rubicon-RNF34 (ZRR) signaling complex that displaced the caspase-1 pseudosubstrate, FliI, from the signaling endosome. The actions of the ZRR complex increased pools of caspase-1 on the signaling endosome that were available for oligomerization. We examine the relevance of ZRR complex-associated signaling in vivo and in patient tissues.

## Results

### Rubicon and RNF34 modulate endosome-associated inflammasome activity

To explore whether ZFYVE21 could regulate oligomerization of inflammasome proteins, we formulated a bioinformatics strategy summarized in Supplementary Fig 1a. The goal of this strategy was to identify protein(s) that could concurrently localize to signaling endosomes and bind to inflammasome protein(s). FYVE domains bind endosome-associated phosphoinositides[19] to mediate homo- and hetero-typic interactions among proteins localized to early

endosomes[20]. NACHT, PYRIN, and CARD domains are variably found on inflammasome components and govern inflammasome oligomerization[13]. We reasoned that proteins dually containing FYVE domains and one or more of the inflammasome-related domains above may localize to early endosomes to regulate oligomerization of inflammasome proteins.

As a first step, we determined the proteome of Rab5-associated inflammasomes. To do this, we adapted our approach for proteomic analyses of MAC+Rab5+vesicles which previously informed the unbiased identification of ZFYVE21 as a Rab5 effector. Three HUVEC lines were stably co-transduced with Rab5-RFP and ASC-GFP and treated with 'high' panel reactive antibody PRA sera, a treatment assembling non-cytolytic MAC pores on ECs[2,3,9–12]. Prior to PRA treatment we pre-exposed HUVECs to IFN-γ for 48–72 h to restore in situ expression of target MHC molecules, thereby allowing optimal alloAb-induced complement activation[10]. We found that, in addition to the above, IFN-γ increased expression of inflammasome components including NLRP3, caspase-1, and gasdermin D in HUVECs, thereby priming NLRP3 inflammasomes[1,2].

Following PRA treatment, ECs were sonicated and Rab5-RFP +vesicles were gated. Large events (Forward Scatter[hi]) showing ASC-GFP fluorescence were subsequently isolated by FACS sorting, subjected to tryptic digestion, and analyzed via LC-MS/MS, using RFP+GFP +Forward Scatter[lo] events as controls (Supplementary Fig 1b). We separated events by size to segregate monomeric vs oligomerized isoforms of inflammasome proteins. Prior to analysis, FACS-sorted vesicles ($n = 168$) were examined by electron microscopy and showed Gaussian size distributions with Forward Scatter[hi]RFP + GFP+vesicles (P2) showing a significantly increased average diameter of $692 \pm 20$ nm vs $89 \pm 18$ in Forward Scatter[lo]RFP + GFP+control vesicles (P1, Supplementary Fig 1c, d). Our workflow revealed 1112 unique proteins appearing in all 3 HUVEC lines of which 402 were upregulated and 473 were downregulated by PRA (Supplementary Fig 1e). As expected, we observed upregulation of proteins annotated to early endosomes (RAB5A) and inflammasomes (PYCARD, CASP1). Molecules appearing in the proteomic analyses above were then queried in an iterative search (Supplementary Fig 1a) involving our prior, unbiased datasets including a proteomic dataset of FACS-sorted MAC+Rab5+ vesicles[9] and a genome-wide siRNA screen performed to identify proteins showing activity against MAC-induced NF-κB[11], an IL-1β-mediated pathway.

We queried a public database (https://prosite.expasy.org/scanprosite/) and generated a list of proteins dually containing FYVE/FYVE-like domains and either NACHT, PYRIN, and/or CARD domains. We did not uncover proteins dually containing FYVE/FYVE-like domains and NACHT or PYRIN (DAPIN) domains, precluding further analysis. However, upon referencing our proteomic datasets leads with proteins dually containing FYVE/FYVE-like domains and CARD/CARD-interacting domains, we uncovered 4 proteins, LRIG3, TCL1A, RNF34, and Rubicon.

We performed validation studies for the 4 proteins above based on parameters specified a priori in our search strategy including localization to early endosomes and the ability to modulate inflammasome activity. Of the 4 candidate proteins, Rubicon and RNF34 levels were induced by PRA (Fig. 1a) and showed increased levels in co-immunoprecipitations (co-IPs) of Rab5 vesicles (Fig. 1b), a population containing MAC+Rab5+endosomes mediating inflammasome activity[1,2]. Rab5 vesicular colocalization of these proteins occurred within 30 min, a timepoint contemporaneous with the appearance of Rab5-associated cleaved caspase-1 in our prior studies. We were unable to detect LRIG3 in Rab5 co-IPs under basal or PRA-treated conditions, and TCL1A, though visualized at high exposures, was not reproducibly upregulated in Rab5 pulldowns with PRA.

We further focused on Rubicon and RNF34. Rubicon is a Rab7 effector[20,21] containing a FYVE-like domain[21] and as such would be

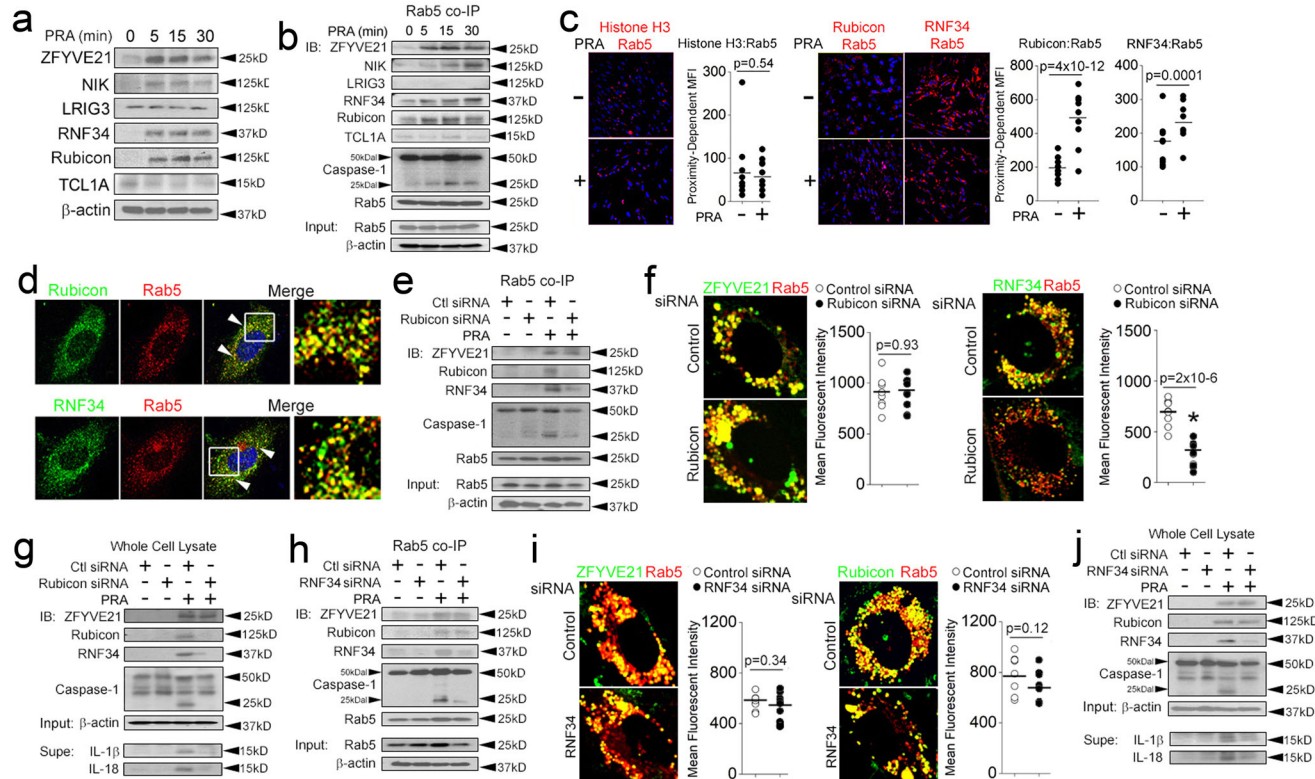

**Fig. 1 | Rubicon and RNF34 Modulate Endosome-Associated Inflammasome Activity.** IFN-γ-pretreated HUVECs were exposed to PRA at the indicated times prior to performing Western blotting (**a**, **g**, **j**) and Rab5 pulldowns (**b**, **e**, **h**). HUVEC stably transduced with Rab5 WT-GFP were treated with PRA for 30 min and subjected to proximity ligation assays (PLAs, **c**). HUVECs were co-transduced with Rubicon-GFP or RNF34-GFP along with Rab5-RFP constructs, treated with PRA for 30 min and analyzed by confocal I.F. (**d**, **f**, **i**). Two-tailed Student's *t*-test (**c**, **f**, **i**) was used for statistical comparisons where *p* < 0.05 is considered statistically significant. Experiments repeated 2 times (**c**, **f**, **i**), 3 times (**b**, **d**, **e**, **g**, **h**, **j**), and 5 times (**a**) with different HUVEC donors. All scale bars, 200 μm. *p*-values are indicated in the figures.

expected to abundantly associate with Rab7+late endosomes. In prior overexpression studies Rubicon, however, was found to colocalize with early endosomes[22,23] in the absence of direct PI(3)P binding but in a manner requiring Rab5 activity[22]. In addition to its roles in macroautophagy[21–25], phagocytosis[21], and endocytosis[26], Rubicon regulates endosome maturation[22,23] and additionally contains a CARD-interacting domain allowing this protein to modulate immune pathways including TLR2 signaling[27] and NF-κB[28]. RNF34/CARP1 is a RING-containing E3 ubiquitin ligase that, like Rubicon, contains multiple functional domains including a FYVE domain, a CARD domain, and a RING domain with E3 ubiquitin ligase activity[29]. Similar to Rubicon, RNF34 localizes to Rab5 vesicles[29], but this localization is mediated by direct binding to PI(3)P[30]. RNF34 participates in inflammatory signaling by mediating degradation of immune-related substrates like caspase-8[31] and MAVS[32] in a manner requiring heterotypic CARD-CARD interactions and its RING domain. Neither protein has been previously connected to inflammasome assembly.

Consistent with co-IP results, we observed that PRA increased the proximity of Rubicon and RNF34 with Rab5 vesicles in proximity ligation assays (PLA, right) but not with a nuclear antigen, histone H3, as a negative control (left, Fig. 1c). To confirm these results, we analyzed Rubicon and RNF34 colocalization with Rab5 using confocal I.F. HUVECs were co-transduced with either Rubicon-GFP or RNF34-GFP along with Rab5-RFP. In these studies, we observed strong colocalization of both Rubicon (Mander's Overlap Coefficient, R = 0.62 ± 0.13) and RNF34 (R = 0.74 ± 0.21) with Rab5 vesicles, consistent with prior reports[22,23,29] (Fig. 1d). Rubicon and RNF34 colocalize on Rab5 endosomes.

Our prior data showed that following PRA treatment caspase-1 activity occurred in association with Rab5 endosomes[2] in a manner requiring ZFYVE21[1] and that endosome-derived cleaved caspase-1 generated the majority of mature IL-1β in ECs[2]. In follow-up studies, we found that Rab5+endosomes containing MAC (C9) also stained for galectin-3, a marker of endomembranous damage (Supplementary Fig 5a), further supporting a role for endosomes as a platform for initiating inflammasome activity. We assessed the functional significance of Rubicon and RNF34 with regards to ZFYVE21 and Rab5-associated inflammasomes. Compatible with our prior studies, in vesicular pulldowns, we observed that PRA elicited caspase-1 activity in association with Rab5 vesicles (Fig. 1b). In Rab5 co-IPs and confocal I.F. studies, we observed that Rubicon siRNA showed no effects on Rab5-associated ZFYVE21, but blocked RNF34 recruitment to Rab5 vesicles (Fig. 1e, f). This was sufficient to decrease Rab5-associated cleaved caspase-1 (25 kD, Fig. 1e). The inability of RNF34 to bind Rab5 vesicles and the attendant loss of Rab5-associated cleaved caspase-1 globally reduced RNF34 and caspase-1 activity in whole cell lysates and blocked generation of mature IL-1β and IL-18 in culture supernatants (Fig. 1g). RNF34 siRNA did not affect levels of Rab5-associated ZFYVE21 or Rubicon (Fig. 1h, i), but reduced endosome-associated caspase-1 activity (Fig. 1h), leading to significantly reduced levels of cleaved caspase-1 in whole cell lysates as well as IL-1β and IL-18 in culture supernatants (Fig. 1j).

Rubicon function has been associated with inhibition of macroautophagy, prompting us to question whether macroautophagy was required to stabilize ZFYVE21 and RNF34. In time course experiments, we found that p62 and LC3-II became rapidly upregulated, indicating that PRA increased autophagic flux (Supplementary Fig 2a). siRNA-

mediated knockdowns of ATG16L or P62 showed no effects on PRA-induced upregulation of ZFYVE21, Rubicon, and RNF34 (Supplementary Fig 2b, c), indicating that autophagic processes did not regulate endosome-bound levels of these proteins. To consolidate the above, we induced non-selective macroautophagy in HUVECs via serum starvation. Phenocopying the above, non-selective macroautophagy did not affect levels of ZFYVE21, Rubicon, and RNF34 in ECs (Supplementary Fig 2d, lanes 1-3). Further, potentiation of autophagic flux with Rubicon siRNA increased generation of LC3-II as previously reported but despite potentiated LC3-II, levels of ZFYVE21 and RNF34 remained unchanged (Supplementary Fig 2d). During non-selective macroautophagy, we found that depletion of P62 with gene-specific siRNA similarly did not affect levels of ZFYVE21, Rubicon, and RNF34 (Supplementary Fig 2e). Corresponding densitometric analyses with statistical comparisons for Fig. 1 were performed and are shown in Supplementary Fig 3. Statistical analyses for densitometries in Supplementary Fig 2 are shown in Supplementary Fig 4. Rubicon and RNF34 localize to early endosomes to modulate Rab5-associated inflammasome activity.

## ZFYVE21 sequesters Rubicon and RNF34 on Rab5 endosomes to enhance their stability

We explored how Rab5 endosomes affected the ability of Rubicon and RNF34 to modulate inflammasome activity. To do this, we stably transduced ECs with Rab5 WT as a control or Rab5 DN (S43N) which is locked in an inactive, GDP-bound state. We found that Rab5 DN displaced Rubicon-GFP and RNF34-GFP from Rab5+vesicles and dispersed Rubicon and RNF34 to small vesicles in the cell periphery lacking Rab5 and that showed weak GFP fluorescence (Fig. 2a, arrowheads). In Rab5 co-IPs, Rab5 DN constructs blocked recruitment of Rubicon and RNF34 to Rab5 vesicles (Fig. 2b). The inability of Rubicon and RNF34 to colocalize to Rab5 vesicles dramatically reduced the stability of these proteins in whole cell lysates to ≤5 min and ablated caspase-1 activity (Fig. 2c). These effects were similarly observed in prior studies involving ZFYVE21 whose colocalization on Rab5 endosomes and whose global protein stability were reduced by loss of Rab5 activity[9]. Rab5 vesicular colocalization mediates the stability of ZFYVE21, Rubicon, and RNF34.

We examined spatial and functional relationships between ZFYE21 with Rubicon and RNF34. In I.F. studies, we detected vesicles staining for ZFYVE21, Rubicon, and RNF34, indicating vesicular apposition of these same proteins (Fig. 2d). We performed gene-specific knockdowns of ZFYVE21 and found that ZFYVE21 siRNA displaced Rubicon and RNF34 from Rab5 endosomes (Fig. 2e, arrowheads) to small, Rab5-negative vesicles in the cell periphery, results similarly observed with Rab5 DN. Loss of Rubicon and RNF34 colocalization with Rab5 vesicles ablated Rab5-associated cleaved caspase-1 (Fig. 2f). These results were phenocopied by PIKIII-mediated depletion of PI(3)P, a phosphoinositide enriched on early endosomes shown previously to act as an obligate lipid binding anchor for ZFYVE21 (Fig. 2g)[9]. These data indicated that Rab5 activity and the Rab5 effector, ZFYVE21, directed Rubicon and RNF34 to Rab5 endosomes to enhance their stability.

In a prior report, we found that ZFYVE21 regulated the stability of NIK, a non-canonical NF-κB mediator[9], and that colocalization of NIK on signaling endosomes, like Rubicon and RNF34, was required for Rab5-associated cleaved caspase-1[1,2]. We thus tested whether NIK and/or caspase-1 affected the stability of Rubicon and RNF34. We found that siRNA-mediated loss of either NIK (Supplementary Fig 5b) or caspase-1 (Supplementary Fig 5c) ablated the generation of cleaved caspase-1 (25 kD) but showed no effect on Rubicon or RNF34 protein levels, indicating that NIK and caspase-1 were both dispensable for mediating the stability of Rubicon and RNF34 on Rab5 endosomes.

We further interrogated mechanisms regulating the stability of Rubicon and RNF34. The rapid increase in Rubicon and RNF34 protein levels within 15–30 min were kinetically consistent with post-translational regulation, a process known to induce NIK. Using NIK and ZFYVE21 as controls, in pulse-chase studies cessation of new protein synthesis with cycloheximide significantly reduced Rubicon and RNF34 levels within 30 min, indicating that these proteins were post-translationally regulated via proteasome degradation (Fig. 2h). To interrogate degradative pathway(s), we inhibited proteasome- or lysosome-mediated degradation with MG132 or leupeptin, respectively, and we used ZFYVE21[9] and STAMBP[33] as respective positive controls for these processes. We observed that Rubicon and RNF34 increased with MG132 (Fig. 2i) but not leupeptin (Fig. 2j) in otherwise untreated samples, and addition of MG132 rescued the ablated stability of Rubicon and RNF34 in ECs transduced with Rab5 DN (Fig. 2k). Corresponding densitometric analyses with statistical comparisons for Fig. 2 are shown in Supplementary Fig 6, and corresponding densitometric analyses with statistical comparisons for Supplementary Fig 5 are shown in Supplementary Fig 4. Taken together, our data are compatible with the interpretation that ZFYVE21 sequesters Rubicon and RNF34 on Rab5 endosomes to shield these proteins from proteasome degradation and that the enhanced stability of Rubicon and RNF34 on Rab5+ endosomes allow these proteins to promote inflammasome activity.

## ZFYVE21, Rubicon, and RNF34 form heterotrimeric complexes

FYVE/FYVE-like and CARD/CARD-interacting domains mediate protein oligomerization, and these domains are variably found on ZFYVE21, Rubicon, and RNF34 which showed vesicular juxtaposition in I.F. studies (Fig. 2d). While we analyzed these proteins based on their abilities to modulate binding among inflammasome components, we considered the possibility that ZFYVE21, Rubicon, and RNF34 might directly interact with each other. In follow-up I.F. studies, Rubicon-GFP and RNF34-RFP showed high punctate colocalization (arrowheads, R = 0.76 ± 0.21, Fig. 3a), indicating that Rubicon and RNF34 were recruited to the same cohort of vesicles. In GFP pulldowns using Rubicon-GFP ECs, Rubicon pulled down RNF34 in a dose-dependent manner (Fig. 3b). We further found that in ZFYVE21-GFP ECs, ZFYVE21 pulled down both Rubicon and RNF34 (Fig. 3c). These data showed inducible interactions among ZFYVE21, Rubicon, and RNF34.

To test direct interactions among these proteins, we performed a series of cell-free assays. Co-incubation of Rubicon and RNF34 proteins resulted in the appearance of a band at ~150 kD (arrowhead, Fig. 3d), and LC-MS/MS analyses of this excised band showed peptides corresponding to Rubicon and RNF34 with 57% and 92% coverage, respectively (Supplementary Fig 5d). These data supported a role for heterodimeric interactions among Rubicon and RNF34. To test interactions of this heterodimeric complex with ZFYVE21, we performed far Western blots using ZFYVE21-GFP as a probe. Compared to GFP probe controls (lane 1, Fig. 3e), the ZFYVE21-GFP probe (~60 kDa) separately bound to both Rubicon (~125 kDa) and RNF34 (~34 kDa) proteins. Binding of the ZFYVE21-GFP probe did not appear to be enhanced when Rubicon and RNF34 proteins were added in combination (lane 4, Fig. 3e). Additionally, ZFYVE21-GFP, when conjugated to agarose beads, separately pulled down Rubicon and RNF34 proteins (Fig. 3f). Co-incubation of Rubicon and RNF34 prior to addition to ZFYVE21-GFP-conjugated beads again did not appear to increase binding among these proteins. In ELISAs, plate-bound ZFYVE21-GFP used as bait showed dose-dependent binding to Rubicon as prey. Reciprocally, we observed that plate-bound Rubicon and RNF34 bound to ZFYVE21 as prey (Fig. 3g). Using 3 cell-free approaches, we observed that ZFYVE21, Rubicon, and RNF34 form a heterotrimeric complex, which we term the 'ZRR' complex.

We investigated the protein domain(s) utilized by Rubicon and RNF34 for binding to ZFYVE21. To do this, we expressed truncation mutants involving the annotated domains of Rubicon in MG132-treated ECs and found that the C-terminal CARD and FYVE-like domains, a region also binding to Rab7[22,23], most strongly regulated

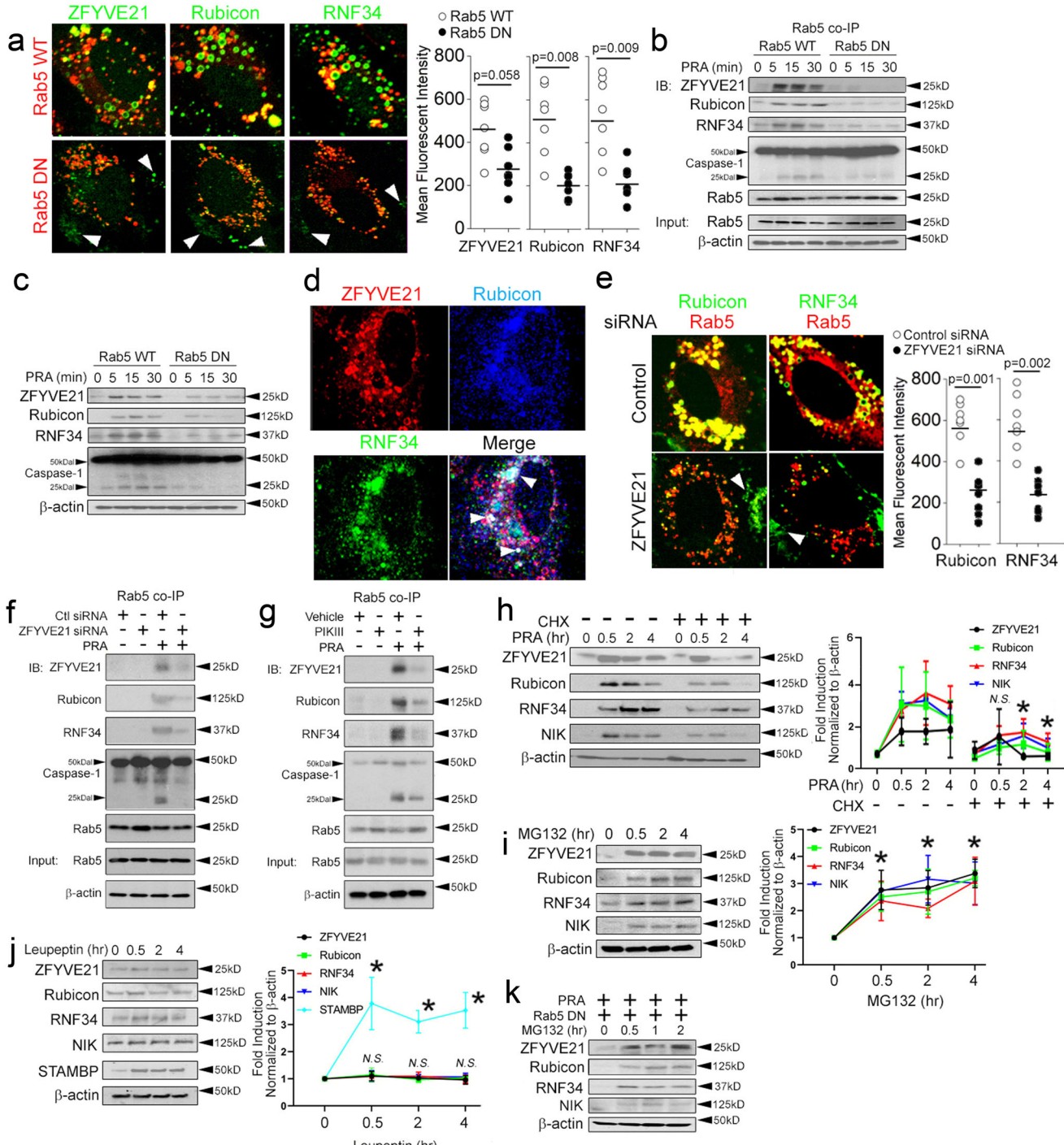

**Fig. 2 | ZFYVE21 sequesters Rubicon and RNF34 on Rab5 endosomes to enhance their stability.** HUVECs stably transduced with Rab5 WT or Rab5 DN were treated with PRA for the indicated times prior to I.F. analyses (**a**), co-IPs (**b**) and Western blotting (**c**). HUVECs were co-transduced with ZFYVE21-RFP, Rubicon-BNP, and RNF34-GFP constructs prior to I.F. analysis (**d**). HUVECs co-transduced with Rubicon and Rab5 or RNF34 and Rab5 constructs were analyzed following control or ZFYVE21 siRNA transfection (**e**). HUVECs were transfected with ZFYVE21 siRNA (**f**) or pre-treated vehicle or PIKIII (5 nM) prior to PRA treatment for 30 min prior to Western blot analysis (**g**). HUVECs were exposed to PRA in the presence or absence of cycloheximide (CHX, **h**, 10 μg/mL), MG132 (**i**, 25 μM), or leupeptin (**j**, 50 μM) for

the indicated times prior to analysis by Western blotting. For Fig. 2h, *p* values involve comparisons of the respective timepoint between treatment with PRA and PRA plus cycloheximide where * indicates *p* < 0.05. For Fig. 2i, j, *p*-values involve comparisons of respective timepoint to timepoint zero where * indicates *p* < 0.05. HUVECs were transduced with Rab5 DN, pre-treated with MG132 and analyzed at the times indicated (**k**). Two-tailed Student's *t* test (**a**, **e**) and one-way ANOVA (Fig. 2h-j) followed by Tukey's pairwise comparison were used for statistical comparisons where *p* < 0.05 is considered statistically significant. Experiments repeated 3 times. All scale bars, 200 μm. Experiments repeated 3 times (**a**–**k**). Data are presented as mean values+/-SD. *p*-values are indicated in the figures.

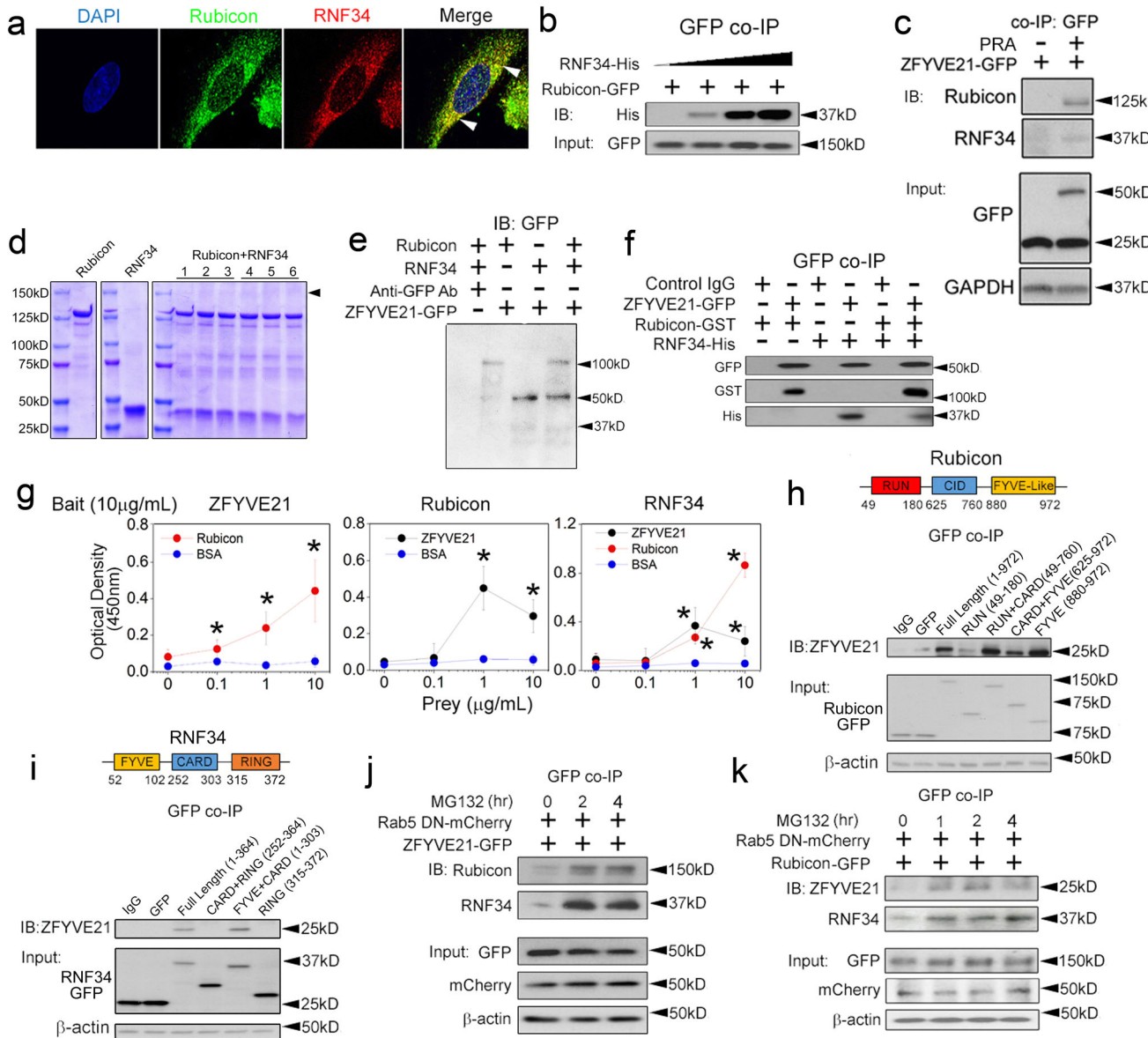

**Fig. 3 | ZFYVE21, Rubicon, and RNF34 form heterotrimeric complexes.** PRA-treated HUVECs were co-transduced with Rubicon-GFP and RNF34-RFP constructs and analyzed by confocal I.F. **a** HUVECs stably transduced with Rubicon-GFP were transiently transfected with increasing concentrations of RNF34-His plasmid prior to GFP co-IPs and Western blotting (**b**). HUVECs stably transduced with ZFYVE21-GFP were treated with PRA for 30 min (**c**). Rubicon (1 μg) and RNF34 (1 μg) were co-incubated at 1 h at room temperature (lanes 1-3) or overnight at 4 °C (lanes 4-6) prior to resolution via SDS-PAGE (**d**). Purified ZFYVE21-GFP (1 μg) was used as a probe using recombinant human Rubicon (1 μg), RNF34 (1 μg), or a 1:1 mixture of Rubicon and RNF34 as bait (**e**). Rubicon-GST (1 μg), RNF34-His (1 μg), or a 1:1 mixture of Rubicon and RNF34 pre-incubated for 4 h at room temperature were added to ZFYVE21-GFP-conjugated agarose beads. Protein complexes were eluted and analyzed via Western blotting under non-denaturing conditions (**f**). Plate-bound ZFYVE21, Rubicon, or RNF34 were incubated with varying concentrations of prey proteins as indicated in ELISAs (**g**). GFP-tagged constructs were overexpressed in HeLa cells following by GFP co-IPs (**h, i**). Rab5 DN-mCherry was stably co-transduced with ZFYVE21-GFP (**j**) or Rubicon-GFP (**k**) in HUVECs and treated with MG132 as indicated prior to GFP co-IPs. *p* values involve relative comparisons with no prey protein added using one-way ANOVA followed by Tukey's pairwise comparison where * indicates $p < 0.05$. Experiments repeated 2 times (**a, b, d–f, j, k**), 3 times (**c, j, k**) or 4 times (**g–i**) with different HUVEC donors. All scale bars, 200 μm. Data are presented as mean values+/-SD. *p*-values are indicated in the figures.

binding to ZFYVE21 (Fig. 3h). In truncation mutant studies of RNF34, AA 52-102, a region containing a FYVE domain, mediated binding to ZFYVE21 (Fig. 3i). Protein regions containing annotated FYVE/FYVE-like domains on Rubicon and RNF34 allowed these proteins to bind ZFYVE21 to permit their sequestration as a complex on Rab5 endosomes. We next asked whether recruitment to Rab5 endosomes was required for ZRR complexes to form. Rab5 DN constructs blocked recruitment of ZRR complex-associated proteins to Rab5 endosomes (Fig. 2b), providing a convenient means to address this question. We co-transduced ZFYVE21-GFP and Rab5 DN-RFP in ECs and pulled down GFP following treatment with MG132. In these cells, we found that

ZFYVE21 inducibly associated with Rubicon and RNF34 (Fig. 3j), a finding we corroborated in reciprocal pulldowns in ECs co-transduced with Rubicon-GFP and Rab5 DN constructs (Fig. 3k). Densitometric analyses with statistical comparisons for Fig. 3 were performed and are shown in Supplementary Fig 7. From this, we concluded that ZFYVE21, Rubicon, and RNF34 may form complexes without requiring Rab5-associated cofactor(s).

## FliI constrains Rab5-associated inflammasome activity

COPs[13] and FliI[15,18] contain CARD/CARD-interacting domains allowing inhibitory pseudosubstrate binding to caspase-1. Based on prior

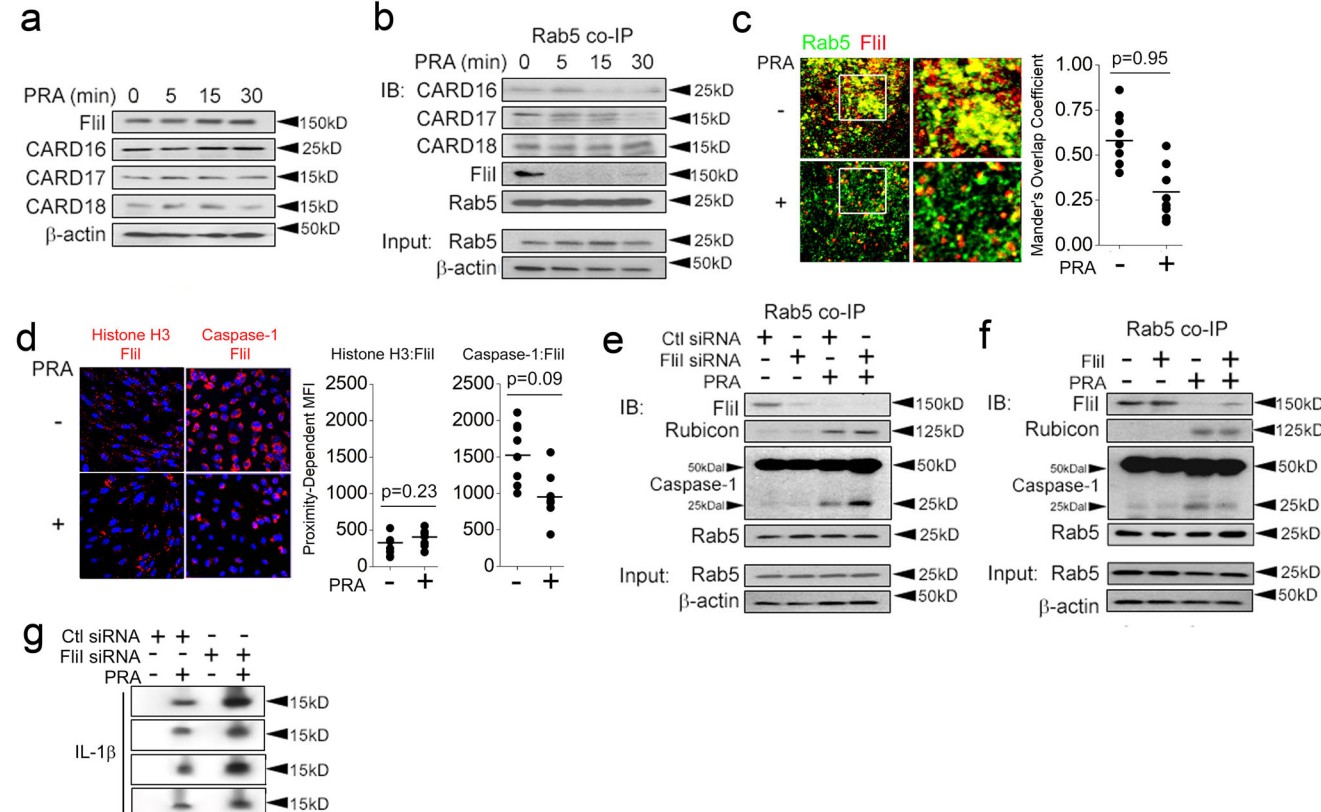

**Fig. 4 | FliI constrains Rab5-associated inflammasome activity.** HUVECs were exposed to PRA at the indicated times and whole cell lysates (**a**) or Rab5 pulldowns (**b**) were analyzed by Western blot. HUVECs were treated for PRA for 30 min and assessed in confocal I.F. (**c**) and PLA (**d**) assays. HUVECs were transiently transfected with FliI siRNA (**e**) or a FliI overexpression construct (**f**) prior to treatment with PRA for 30 min in Western blot studies. HUVECs were transfected with control or FliI siRNA and treated for 4 h prior to analyses of culture supernatants (**g**). Two-tailed Student's *t*-test was used for statistical comparisons (**c**, **d**) where *p* < 0.05 is considered statistically significant. Experiments repeated 3 times (**a**–**g**) with different HUVEC donors. All scale bars, 200 μm. *p*-values are indicated in the figures.

studies showing that the effector function of Rubicon is reliant on its ability to form novel heterotypic CARD-CARD interactions[24,25,27,28] we hypothesized that the Rubicon component of the ZRR complex could interact with COPs and/or FliI in a manner that could destabilize their inhibitory interactions with caspase-1. To explore this, we initially characterized the distribution of caspase-1 on signaling endosomes. With PRA, caspase-1 inducibly associated with NLRP3 but no other inflammasome sensors including NOD1, AIM2, or NLRC4, indicating that PRA treatment solitary assembled NLRP3 inflammasomes (Supplementary Fig 5e). In analyses involving the same samples, we dually detected both inflammasome complexes and ZRR complexes in association with Rab5 (Supplementary Fig 5 f). The Rab5+ vesicular population was found to be heterogeneous and was comprised of Rab5+Caspase-1+ vesicles containing (top, Supplementary Fig 5 g) and lacking MAC (bottom). These results showed that PRA treatment of ECs solitarily forms NLRP3 inflammasomes and that the caspase-1 component of these NLRP3 inflammasomes shows a heterogeneous vesicular distribution with respect to MAC+Rab5+endosomes.

We proceeded to analyze caspase-1-binding pseudosubstrates including COPs (CARD16, CARD17, and CARD18) and Flightless I (FliI) in ECs. We did not observe changes in pseudosubstrate levels in whole cell lysates following PRA treatment (Fig. 4a). However, in pulldowns of Rab5 vesicles, sites where inflammasome activity occurs, we observed that the inhibitory pseudosubstrates, CARD16, CARD17, and FliI, were removed from Rab5 vesicles to varying degrees within ≤30 min (Fig. 4b). We did not detect significant changes in CARD18 levels in Rab5 co-IPs with PRA. Due to its high basal expression on Rab5 vesicles as well as its robust and reproducible loss with PRA treatment, we focused more closely on FliI and observed loss of proximity of this

molecule with Rab5 vesicles in I.F. (Fig. 4c) and PLA studies (Fig. 4d). To test a role for FliI in Rab5-associated inflammasomes, we performed loss- and gain-of-function studies involving FliI. Consistent with the above, PRA treatment in the presence of control siRNA caused loss of FliI on Rab5 vesicles, and this occurred concurrently with the appearance of Rab5-associated Rubicon and cleaved caspase-1 isoforms (lane 1 *vs* lane 3, Fig. 4e). Conversely, overexpression of full-length FliI, whose N- and C-terminal domains interact with caspase-1[15], reduced Rab5-associated cleaved caspase-1 (Fig. 4f, lane 3 *vs* lane 4). Functionally, FliI siRNA potentiated the generation of elaborated IL-1β, confirming its described role as an endogenous inhibitor of inflammasome activity (Fig. 4g). Corresponding densitometric analyses with statistical comparisons for Fig. 4 were performed and are shown in Supplementary Fig 8. These data show that FliI, a caspase-1 pseudosubstrate, constrains inflammasome activity and is inducibly removed from Rab5 endosomes, sites where caspase-1 activity occurs.

### Rubicon and RNF34 cooperatively reduce the stability of Rab5-associated FliI

Based on the reasoning detailed above, we tested whether Rubicon was capable of destabilizing inhibitory caspase-1:FliI interactions. In reciprocal co-IPs, monomeric, i.e., uncleaved, procaspase-1 (50 kD) showed strong associations with FliI under basal conditions, a time-point showing minimal inflammasome activity (Fig. 5a, b). With PRA, this basal interaction between procaspase-1 and FliI was lost, and FliI rapidly shifted its binding to Rubicon which was complexed with RNF34 (Fig. 5a, b). The observed change in FliI binding interactions occurred antecedent to the appearance of caspase-1 activity. We performed truncation mutant studies of Rubicon to map relevant

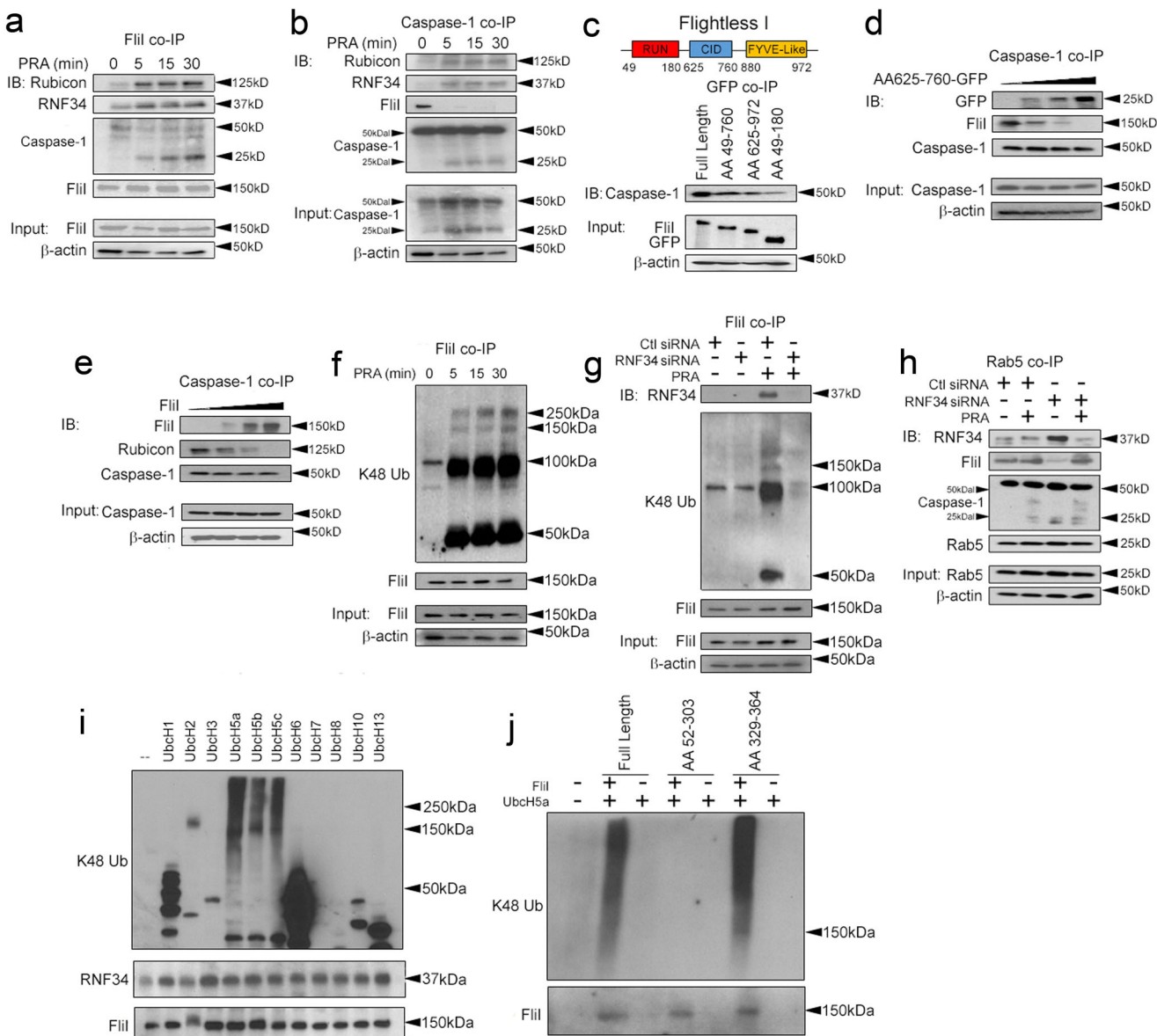

**Fig. 5 | Rubicon and RNF34 cooperatively reduce the stability of Rab5-associated fliI.** HUVECs were exposed to PRA for the indicated times prior to performing pulldowns for FliI (**a**) and caspase-1 (**b**). HeLa cells were transiently transfected with various GFP-tagged overexpression vectors prior to GFP co-IPs (**c**). HUVECs were transiently transfected with increasing plasmid concentrations encoding AA 625-760 of Rubicon fused with GFP, (**d**) or full-length FliI (**e**) followed by pulldowns for caspase-1. HUVECs stably transduced with Ubiquitin WT-HA (Ub-HA) were exposed to PRA for the indicated times followed by pulldowns for FliI (**f**).

Ub-HA HUVECs were transfected with control or RNF34 siRNA as indicated and treated with PRA for 30 min (**g**). HUVECs were transfected with siRNA as indicated followed by Rab5 co-IPs following PRA treatment for 30 min (**h**). In ubiquitin ligase assays, recombinant human FliI (0.5 μg) was incubated as substrate with activated Ub, the indicated E2 ligase(s), and native RNF34 protein (0.5 μg, **i**) or RNF34 truncation mutants prior to Western blotting (**j**). Experiments repeated 3 times with different HUVEC donors.

domain(s) binding to procaspase-1 and found that AA 625–760, shown previously to be a CARD-interacting domain[28], showed strong binding to procaspase-1 (50 kD). These data together suggested competitive binding interactions between Rubicon and FliI, potentially involving AA 625–760. To formally test this, we overexpressed AA 625–760 of Rubicon and in a dose-dependent manner, we observed decreased binding between FliI and procaspase-1 (Fig. 5d). Reciprocally, dose-dependent overexpression of full-length FliI blocked the ability of Rubicon to associate with procaspase-1 (Fig. 5e). Following the observation that Rubicon could block interactions between FliI and caspase-1, we further phenotyped interactions of Rubicon along with ZFYVE21 and RNF34 with other components of the canonical inflammasome. In ELISAs, ZRR complex proteins did not show appreciable binding to

NLRP3 which lacks a CARD domain (Supplementary Fig 5h). However, among ZRR complex proteins we observed that Rubicon showed reciprocal binding to ASC at the concentrations tested (Supplementary Fig 5h), further supporting the notion that ZRR complexes interact with inflammasomes. Our data indicated that the CARD interacting domain of Rubicon competitively displaces inhibitory binding between FliI and procaspase-1.

RNF34 displays E3 ubiquitin ligase activity and modulates the stability of immune-related proteins via K48-ubiquitinylation[29–32]. Due to its ability to bind Rubicon (Fig. 3) and its enrichment on Rab5 endosomes (Fig. 1), a compartment where FliI was inducibly removed (Fig. 4), we examined whether RNF34 could regulate the stability of FliI on signaling endosomes. Such regulation would be teleologically

sensible to prevent liberated pools of FliI from competitively re-associating with caspase-1. In ubiquitin-HA ECs, we observed that FliI became rapidly K48 ubiquitinylated following PRA treatment within 5 min (Fig. 5f), a process kinetically compatible with the observed removal of FliI from Rab5 endosomes (Fig. 4). In FliI pulldowns, we observed non-discrete ubiquitin signals exceeding ≥250 kDa, consistent with polyubiquitinylation of FliI by RNF34. Additionally, we consistently observed a more discrete band at ~150 kDa possibly representing monoubiquitnylated isoforms of FliI. Gene-specific knockdowns of RNF34 reduced Fli1 ubiquitinylation (Fig. 5g) and restored the observed PRA-induced loss of FliI from Rab5 vesicles (Fig. 5h). These processes had the overall effect of attenuating Rab5-associated caspase-1 activity (Fig. 5h). To test whether RNF34 could directly ubiquitinylate Fli1, we performed a series of cell-free ubiquitinylation reactions. In vitro pilot studies revealed that the E2 conjugating enzymes, UBCH5a, UBCH5b, and UBCH5c, mediated polyubiquitinylation of FliI by RNF34 (Fig. 5i). In these reactions, we again observed non-discrete and semi-discrete bands at >250 kDa and ~150 kDa, respectively. Due to its strong activity, we used UBCH5a as an E2 conjugating enzyme and found that RNF34 directly mediated K48 ubiquitinylation of FliI, and loss of its RING domain abrogated this effect (Fig. 5j). Upon reprobing, we did not observe K63 ubiquitinylation of FliI by RNF34. These data showed that RNF34 directly mediates K48 polyubiquitinylation and possibly monoubiquitinylation of FliI. Corresponding densitometric analyses with statistical comparisons for Fig. 5 were performed and are shown in Supplementary Fig 9. Altogether, Rubicon and RNF34 cooperatively remove FliI from Rab5 endosomes to increase pools of endosome-bound caspase-1 available for.

## ZRR complexes form in patient tissues

We assessed the clinical relevance of ZRR complexes, initially asking whether ZRR complexes could form in human tissues. To address this question, we tested whether PRA could assemble ZRR complexes in human cortical kidney tissues in organ culture. We chose to analyze this compartment due to its high EC content occurring in glomerular and peritubular capillary beds. In pilot studies, we found that cortical regions of human kidneys responded to PRA after treatment for 4 h and showed evidence of MAC (arrowheads, C9) deposition on glomerular ECs (Ulex, Supplementary Fig 10a). In 3 separate donors, ZFYVE21 levels appeared similar in lysate inputs in both untreated and PRA-treated samples, which we surmise was due to basal expression of ZFYVE21 in glomerular ECs[9]. Exploiting this, we performed ZFYVE21 co-IPs and found that ZFYVE21 pulled down both Rubicon and RNF34 with PRA treatment (Fig. 6a). To assess whether ZRR complexes formed in situ, we analyzed cortical kidney tissues from healthy control subjects (HC, n = 6) or subjects with antibody-mediated rejection (ABMR, n = 6). We found that ZFYVE21 pulled down Rubicon and RNF34 in ABMR tissues but not in HC tissues (Fig. 6b). We performed dual immune-EM of ABMR tissue and found close apposition of ZFYVE21 (arrows) with Rubicon (arrowheads, top) on intracellular vesicles (Fig. 6c), suggesting ZRR complex formation on endosomes in vivo. Based on these data, we concluded that ZRR complexes are capable of forming in human tissues and that ZRR complexes are increased in ABMR, a complement-mediated disease condition.

We next performed I.F. analyses of Rubicon in various complement-mediated conditions including coronary artery tissues from heart transplant patients with ABMR (n = 6, Fig. 6d), synovial tissues from patients with rheumatoid arthritis (RA, n = 4, Fig. 6e), and renal tissues from patients with systemic lupus erythematosus (SLE, n = 5, Fig. 6f). We used ZFYVE21, shown previously to highly co-localize with ECs in these same cohorts[9], as a positive control. In FFPE sections in which the above biopsies were obtained, we were unable to obtain specific RNF34 staining, precluding analysis of this molecule in these biopsies. Across cohorts, we found that Rubicon was expressed at low levels in control biopsies and was upregulated in chronic ABMR (CABMR), SLE, and RA in regions colocalizing with ECs (Ulex) bound by complement (C4d, C6, Fig. 6d–f). In CABMR and RA tissues, Rubicon, whose pixel overlap with Ulex+ECs occurred at similar levels to that of ZFYVE21 (Fig. 6d, e), also showed punctate staining in regions not co-localizing with Ulex, possibly reflecting expression in immune cells (arrows, Fig. 6d, e). In control renal tissues, as previously[9] we observed high basal levels of glomerular ZFYVE21, but we did not similarly observe glomerular Rubicon staining in control patients (Fig. 6f). In SLE renal biopsies, Rubicon heavily co-localized with both glomerular (arrows) and peritubular capillary beds (arrowheads, Fig. 6f). In parallel to the above, we retrieved RNA seq datasets from the Gene Expression Omnibus and assessed correlations among the gene abundances of ZFYVE21, Rubicon, and RNF34 in kidney tissues from renal transplant patients with CABMR (n = 110), synovial tissues from patients with RA (n = 23), and kidney tissues from patients with SLE nephritis (n = 21). We found that ZFYVE21, Rubicon, and RNF34 gene abundances showed high correlations with each other (Fig. 6g) and showed moderate correlations with an index of NF-κB-dependent genes reflecting EC activation (VCAM1, ICAM1, and SELE, Fig. 6h). To support these findings, we performed correlative I.F. staining of HC and ABMR renal biopsies. HC biopsies as observed previously showed basal staining for ZFYVE21 in glomeruli, and MFIs for ZFYVE21 + CD31+Rab5+ECs became increased in ABMR biopsies (Supplementary Fig 5i). In HC biopsies, we observed negligible basal staining for Rubicon in peritubular capillaries, and MFIs for Rubicon+CD31+Rab5+ECs became dramatically increased in ABMR samples. Together, these data show that ZRR complex-associated proteins are upregulated in patient ECs during complement-mediated disease.

## ZRR complex-associated signaling occurs in vivo

We next tested whether salient steps involving ZRR complex formation occur in vivo, employing 3 separate mouse models. In a human artery xenograft model, human coronary artery segments were surgically implanted as interposition xenografts in the infrarenal aortic position of immunodeficient SCID/beige mice who were subsequently injected with PRA. Twenty-four hours later grafts were harvested and analyzed by I.F. Using this model, we previously found that ZFYVE21 became upregulated primarily in intimal ECs with scant punctate staining occurring in the adventitia[9]. With PRA, we found that Rubicon and RNF34 became upregulated and became juxtaposed in the intima in PLA studies in vivo (n = 3 per group, Fig. 7a). These effects occurred in association with increased sera levels for human IL-1β and IL-18 (Fig. 7b). In a second approach, we employed a collagen gel-fibronectin gel model which allowed us to ask whether ZRR complex-associated signaling occurred in vivo. HUVECs suspended in gel matrices were implanted subcutaneously into flanks of SCID/bg hosts. Three weeks post-implantation, hosts were injected with PRA, and gels were harvested 24 h later and analyzed by I.F. PRA-treated gels showed complement activation (C4d) occurring on human ECs (Ulex) in association with E-selectin, a marker for EC activation (Supplementary Fig 10b, c). Prior to implantation, we transduced HUVECs with Rab5 WT or Rab5 DN constructs and found that, while complement activation on ECs was unchanged in Rab5 WT vs Rab5 DN gels, ZFYVE21 was significantly reduced in Rab5 DN gels, in accord with our prior results[9] (Supplementary Fig 10d, e). HUVECs stably transduced with Rab5 DN constructs additionally showed decreased MFIs for both Rubicon (top, Fig. 7c) and RNF34 (bottom), indicating that Rab5 activity was required for stabilizing these ZRR complex components in ECs in vivo. We next analyzed the requirement for Rubicon and RNF34 for PRA-induced inflammasome activity. Compared to ECs transduced with control shRNA, we observed that shRNA against either Rubicon (Fig. 7d) or RNF34 (Fig. 7e) decreased levels of endothelial cleaved caspase-1, and these changes were associated with decreased human IL-1β and IL-18 protein in sera (Fig. 7f). These data indicated that ZRR complex

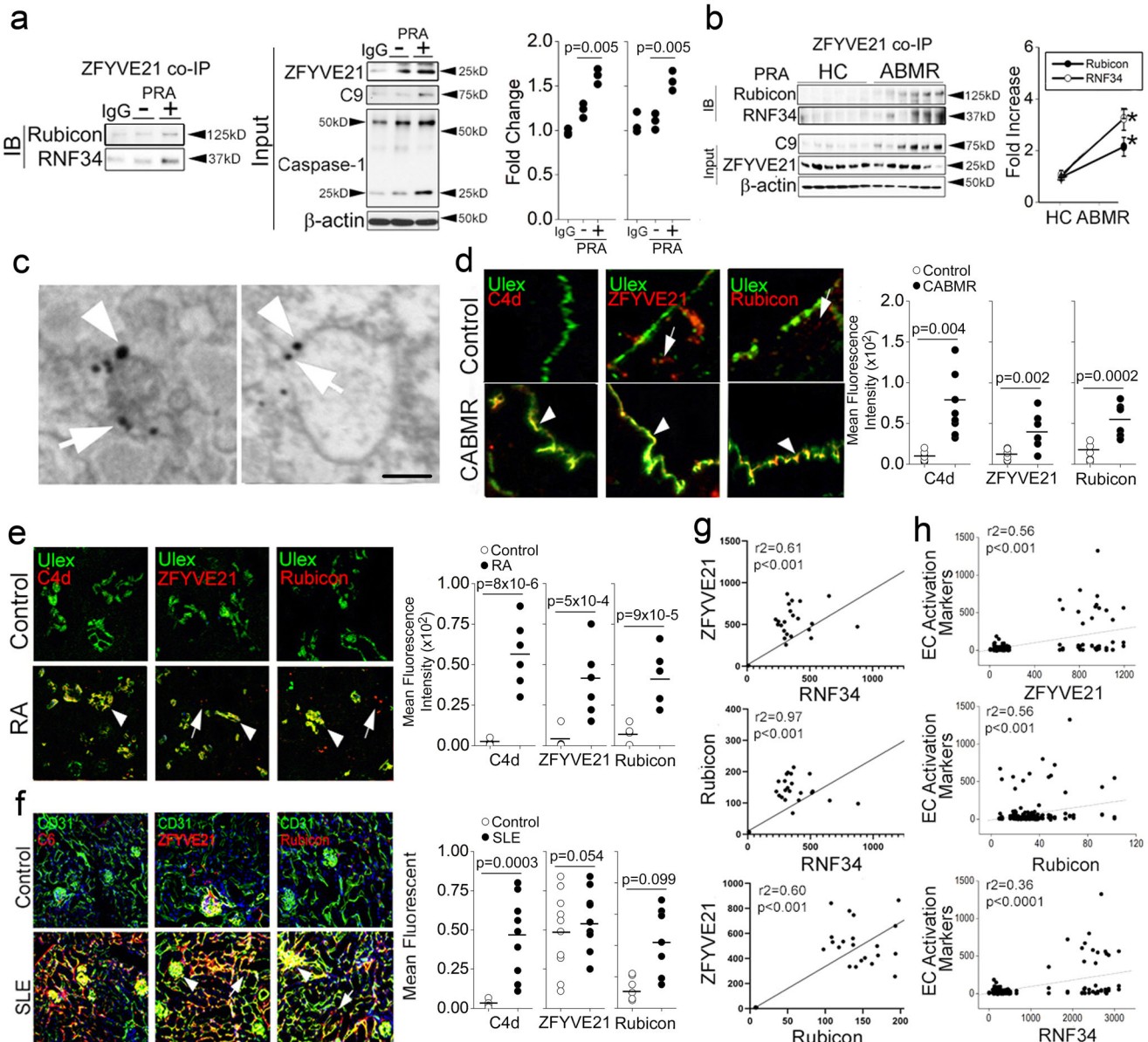

**Fig. 6 | ZRR complexes form in patient tissues.** Cortical renal tissue from control patients were placed in organ culture and treated with PRA for 4 h prior to ZFYVE21 co-IP and Western blot analysis (**a**, n = 3). Cortical renal tissues from healthy control (HC) or antibody-mediated rejection (ABMR) were analyzed by co-IP (n = 6 per group, **b**). ABMR tissues were analyzed by dual immune-electron microscopy for ZFYVE21 (arrowheads, n = 3, **c**) and Rubicon (arrows). FFPE biopsies from patients with CABMR (n = 6, **d**), RA (n = 4, **e**), or SLE (n = 5, **f**) were stained and analyzed as indicated. Transcriptomic data from renal tissues from renal transplant patients with CABMR (n = 110), synovial tissues from patients with RA (n = 23), and renal tissues from patients with SLE nephritis (n = 21) were obtained from the Gene Expression Omnibus. One-way ANOVA with Tukey's post-hoc comparison (**a**), Two-tailed Student's t test (**b**, **d**–**f**), and Pearson's correlations (**g**, **h**) were used for statistical comparisons where p < 0.05 is considered statistically significant. All scale bars, 400 μm. Data are presented as mean values+/-SD. p-values are indicated in the figures. All scale bars, 200 μm.

proteins became upregulated in a Rab5-dependent manner and that ZRR complexes were required to regulate inflammasome activity in human vessels in vivo.

We supported the results above using a genetic approach. In four treatment groups with n = 3 per group, WT or Rubicon-/- mice (H-2$^b$) were injected with isotype control (MOPC) Ab or anti-H-2$^b$ Ab to elicit systemic Ab-induced complement activation. Following anti-H-2$^b$ Ab injection, twenty-four hours later renal peritubular capillaries showed that MAC (C6) deposition had specifically occurred on ECs in the absence of significant TUNEL staining, indicating that anti-H2$^b$ Ab had elicited non-cytolytic MAC assembly on ECs (Supplementary Fig 10f), and subsequent to this validation we harvested kidney tissue for

biochemical studies. Compared to control hosts treated with MOPC Ab, hosts treated with anti-H-2$^b$ Ab showed ZRR complex recruitment in Rab5 pulldowns (Fig. 7h), and ZRR complex recruitment occurred in association with significantly decreased Rab5-associated FliI and increased Rab5-associated cleaved caspase-1 (Fig. 7h). In anti-H2$^b$-treated Rubicon-/-hosts, Rubicon deficiency decreased recruitment of RNF34 but not ZFYVE21 to Rab5 vesicles, recapitulating our in vitro findings in Fig. 1. Rubicon deficiency moreover restored Rab5-associated FliI and attenuated the generation of cleaved caspase-1 as observed in HUVECs in Fig. 4. Densitometric analyses with statistical comparisons for this experiment are shown in Supplementary Fig 9f. The original, uncropped Western blot films pertinent to the analyses

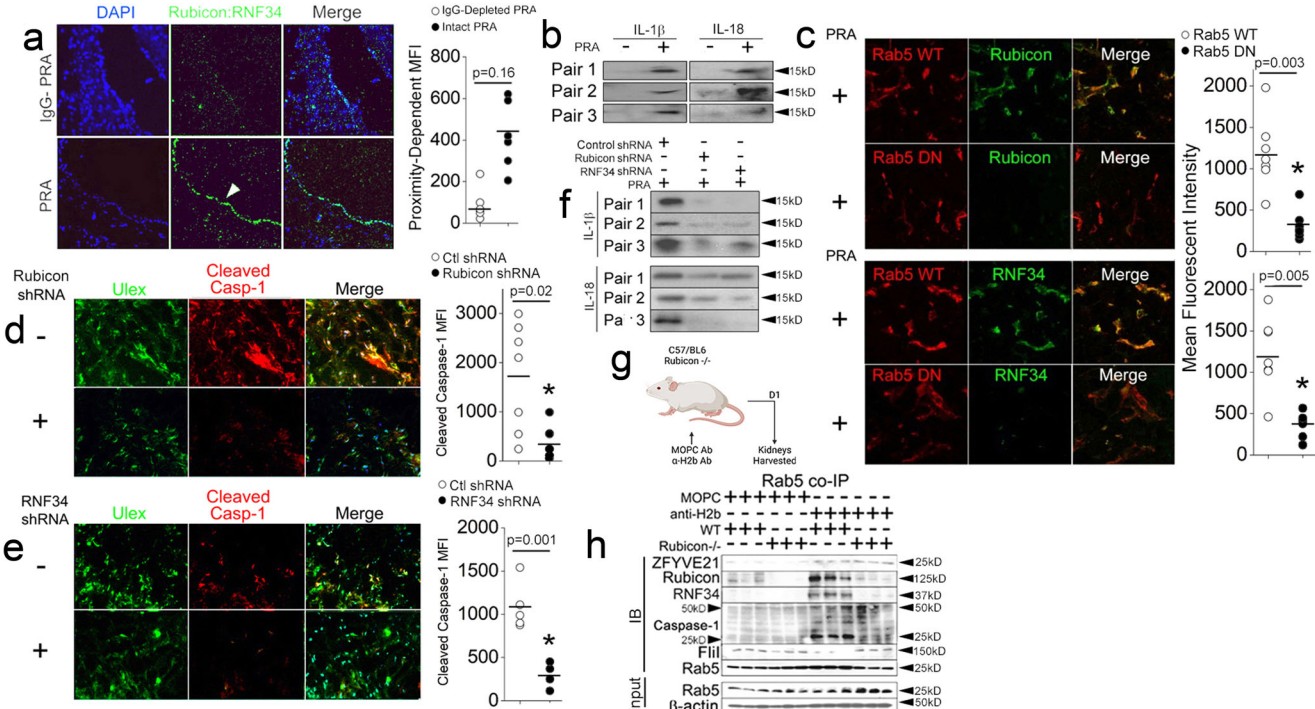

**Fig. 7 | ZRR complex-associated signaling occurs in vivo.** Human artery xenografts implanted in immunodeficient SCID/beige hosts were exposed to PRA for 24 h (200 μL i.v. tail vein injection per mouse) and analyzed in proximity ligation assays (**a**, $n = 3$ per group). Sera from SCID/beige hosts bearing PRA-treated human artery xenografts were analyzed by Western blot (**b**, $n = 3$ per group). HUVECs stably transduced with Rab5 WT mCherry or Rab5 DN mCherry (**c**), Rubicon shRNA (**d**) or RNF34 shRNA (**e**) were embedded in collagen gel matrices, implanted in SCID/beige mice for 2–4 weeks, and microvessels were exposed to PRA for 24 h (200 μL i.v. tail vein injection per mouse) prior to staining and analysis by I.F ($n = 3$ per group). Sera from SCID/beige hosts implanted with collagen gel matrices were analyzed by Western blot (**f**, $n = 3$ per group). C57/Bl6 or Rubicon-/-mice were injected with 750 μg MOPC or anti-H-2$^b$ Abs (**g**), and kidneys were harvested for Rab5 co-IPs 24 h later ($n = 3$ per group, **h**). Figure 7g was created with BioRender.com. Two-tailed Student's $t$-test was used for statistical comparisons (**a**, **c**–**e**) where $p < 0.05$ is considered statistically significant. Experiments repeated 3 times. All scale bars, 400 μm. $p$-values are indicated in the figures.

above as well as those throughout the manuscript are provided in Supplementary Fig 11–20. Using three mouse models, we show that ZRR complex-associated signaling occurs in vivo.

### ZRR complexes regulate complement-induced vascular injury in vivo

We tested effects of ZRR complexes on EC activation. We previously showed that ZFYVE21 induced assembly of NLRP3 inflammasomes, causing IL-1β-dependent activation of canonical NF-κB and EC activation in vitro and in vivo[2,3]. In NF-κB luciferase HUVECs, NF-κB activity was strongly induced by PRA as previously observed[11]. We found that siRNA vs Rubicon or RNF34 significantly reduced NF-κB activity relative to ECs transfected with control siRNA (Fig. 8a). Orthogonally, nuclear translocation of P65, part of the terminal heterodimer activating canonical NF-κB, was significantly reduced by siRNA against Rubicon and RNF34 relative to control siRNA (Fig. 8b). In alignment with the above, Rubicon siRNA and RNF34 siRNA significantly reduced transcripts for a panel of NF-κB-dependent genes upregulated by PRA[10] and shown previously to require IL-1β[2] (Fig. 8c). We then examined effects of ZRR complexes on EC immunogenicity using EC:T cell cocultures. Human ECs, unlike those from murine hosts, are capable of directly stimulating allogeneic CD4+ memory T cells (Tmem) that go to elaborate IFN-γ, a vasculopathic cytokine[34,35]. IFN-γ responses by Tmem are potentiated by IL-1β[6], and accordingly, vasculopathic lesions in a humanized mouse model are enhanced by IL-1β and IL-18-producing ECs in response to MAC[1–3]. Phenocopying effects previously observed for ZFYVE21[9], we found that gene-specific knockdowns for Rubicon or RNF34 significantly reduced frequencies of IFN-γ+ Tmem (Fig. 8d). ZRR complexes modulate NF-κB activity to elicit EC activation.

Based on the data above, we proceeded to examine effects of ZRR complexes on alloimmune tissue injury, employing a model of chronic rejection against the H-Y alloantigen, a minor mismatch allele. Male WT or Rubicon-/-mice were treated with anti-H-2$^b$ Ab to elicit MAC assembly on ECs within skin allografts which were then placed onto recipient SCID/bg hosts that had passively received female splenocytes (Fig. 8e, $n = 8$ mice per group). Prior to implantation, we analyzed skin grafts for complement activity and EC death (Supplementary Fig 10f). Compared to anti-H2$^b$-treated WT grafts, anti-H2$^b$-treated Rubicon-/- grafts showed decreased glomerular VCAM-1 staining (Fig. 8f). These data indicated that anti-H2$^b$ treatment elicited non-cytolytic MAC deposition on ECs to cause Rubicon-dependent EC activation. Three weeks following skin implantation, we found that WT grafts receiving anti-H-2$^b$ Ab developed strong perivascular CD3+ infiltration which was significantly attenuated in similarly-treated Rubicon-/-grafts (Fig. 8g). Epidermal hyperplasia is an IFN-γ-mediated phenotype indicative of tissue injury[36] whose severity is enhanced by IL-1 family cytokines[37]. Compared to MOPC-treated WT controls, we observed increased epidermal thickening in anti-H-2$^b$ Ab-treated WT grafts, and this phenotype was significantly attenuated in similarly-treated Rubicon-/- grafts (Fig. 8h). Following complement activation on ECs, Rubicon, a component of the ZRR complex, regulates perivascular inflammation and tissue injury in vivo.

## Discussion

Informed by a bioinformatics strategy, we identify a ZFYVE21-Rubicon-RNF34 (ZRR) signaling complex promoting inflammasome activity on endosomes. Nascent signaling endosomes basally contain caspase-1 bound to its inhibitory pseudosubstrate, FliI, and thus show limited capacities for inflammasome activity (Fig. 8i, I). Following transfer of MAC to Rab5 endosomes, ZFYVE21 elicits endosome sequestration of

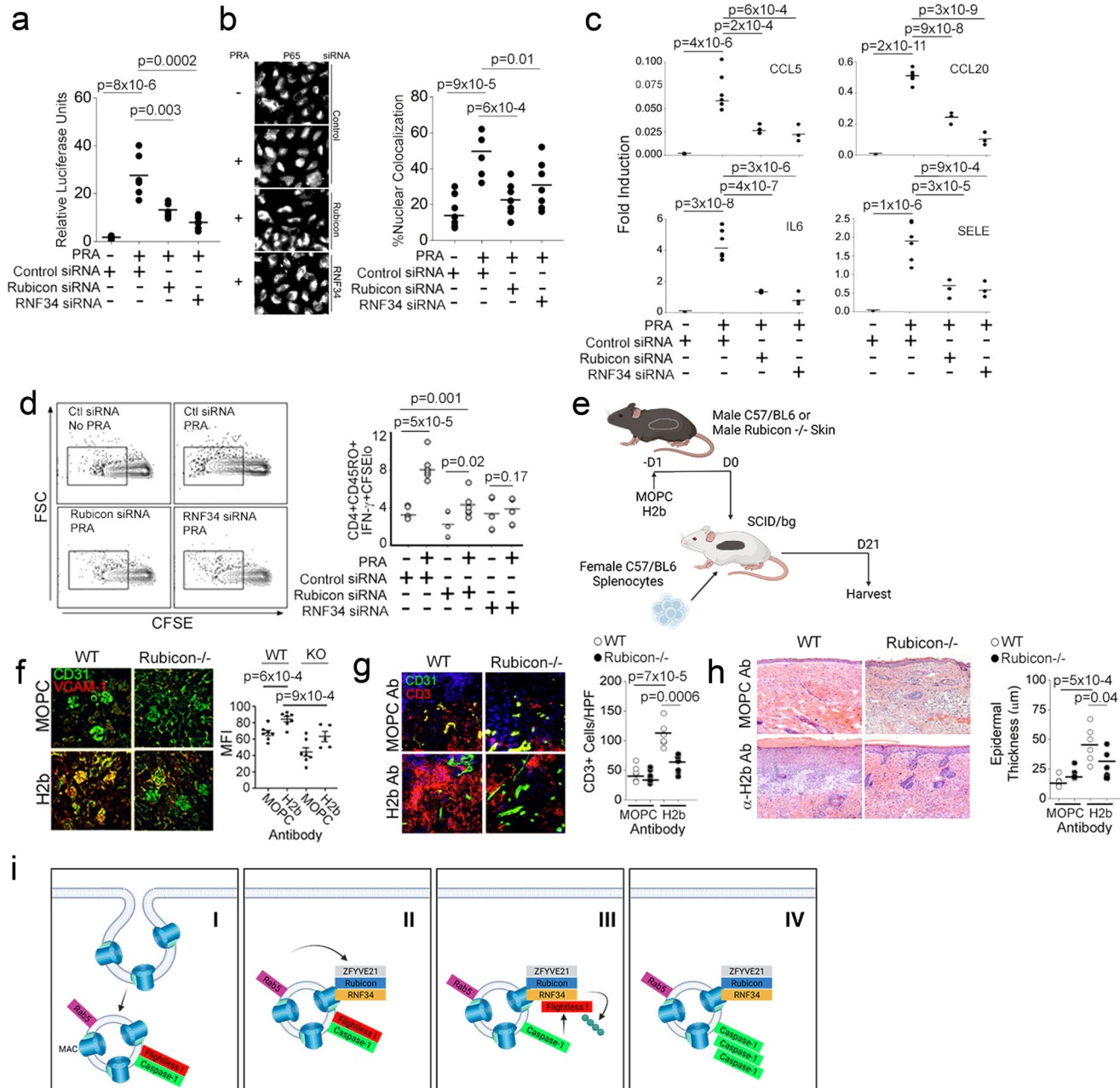

**Fig. 8 | ZRR complexes regulate complement-induced vascular injury in vivo.** HUVECs stably transduced with an NF-κB luciferase reporter were treated with PRA and luciferase activity were assessed following treatment with PRA for 6 h (**a**). Quantitative epifluorescence analysis of nuclear translocation of the P65 NF-κB subunit in HUVECs transfected with siRNA and treated with PRA for 4 h (**b**). qRT-PCR of siRNA-transfected HUVECs exposed to PRA for 4 h (**c**). EC:T cell cocultures were performed using HUVECs transfected with various siRNA as indicated, and CFSE-labeled T cells were harvested and analyzed by FACS after 10–14 days in coculture (**d**). Male WT or Rubicon-/-mice received 200 μL tail vein injection of MOPC or anti-H-2b Ab (750 μg) and 24 h later, male skin grafts were harvested and analyzed by I.F. implanted onto female SCID/bg hosts that had passively received $1 \times 10^6$ female WT C57/Bl6 splenocytes. Three weeks post-implantation, skin grafts

were analyzed (**e**). Following MOPC or anti-H-2b Ab injection i.v. (750 μg), kidney tissues from age-matched WT or Rubicon[-/-] mice were harvested 24 h later and analyzed for glomerular VCAM-1 staining on CD31+ECs prior to host implantation (**f**, n = 8 mice per group). Three weeks post-implantation, male skin grafts treated with MOPC or anti-H-2b Ab as above were analyzed by I.F. (**g**, n = 8 mice per group) and H&E sections (**h**, n = 8 mice per group). Schema of the proposed mechanism for ZRR complexes (**i**). For Fig. 8a–d, experiments repeated 3 times with different HUVEC donors. For Fig. 8a–d, f–h two-way ANOVA followed by Tukey's pairwise comparison was used for statistical analysis where p < 0.05 is considered statistically significant. Figures 8e and 8i were created with BioRender.com. All scale bars, 400 μm. p-values are indicated in the figures.

Rubicon and RNF34, shielding these proteins from proteasome degradation. This process post-translationally enhances the stability of ZRR complexes on signaling endosomes (II). The cooperative actions of Rubicon and RNF34 remove the FliI pseudosubstrate from the signaling endosome (III). Loss of FliI increases unbound pools of Rab5-associated caspase-1 available for activation (IV). Conceptually, we

provide evidence that the signaling endosome enhances the stability of signaling proteins but also promotes their spatial juxtaposition to allow formation of newly functional protein complexes. These complexes include ZRR complexes and NLRP3 inflammasomes. The dual functions of the signaling endosome functionally induces EC activation via IL-1β-mediated NF-κB.

An antecedent priming step upregulates and licenses inflammasome proteins for assembly. While induction of inflammasome proteins canonically occurs following PAMP-induced NF-κB, this process may be subsumed by endogenous molecules including reactive oxygen species, metabolites, and hypoxia during sterile inflammation associated with transplant rejection[38]. Various cytokines including IFN-γ, though dispensable for IL-1β production in human macrophages, may similarly prime NLRP3 inflammasomes[39], a process that we found occurs following EC-mediated direct allorecognition in vitro and in various humanized mouse models including coronary artery xenografts[2,9,40] in vivo. While abundant IFN-γ is induced in our in vitro and in vivo approaches above, our studies do not rule out the contribution(s) from other endogenous molecules eliciting inflammasome priming. Our data herein using murine and humanized models show that in situ levels of endogenous priming molecule(s) are sufficient for generating IL-1β by ECs in vivo in the absence of an experimentally-induced priming step. As the ZRR complex is newly discovered, defining such signals with regards to the function of this complex will be required to fully define the scope of its function(s) in vivo and is the focus of ongoing studies by our group.

Licensing of NLRP3 inflammasomes involve post-translational modifications including tyrosine phosphorylation and ubiquitinylation that regulate the ability of NLRP3 to oligomerize[15–17]. Our data show that NLRP3 is inducibly recruited to signaling endosomes by PRA, and as such, subcellular trafficking of NLRP3 may confer an additional layer of inflammasome regulation. We as of yet do not understand how signaling endosomes communicate with surrounding organelles to allow inflammasome biosynthesis. Elucidation of relevant trafficking processes may uncover druggable targets to attenuate terminal inflammasome activity at the licensing step involving protein enrichment on signaling endosomes.

Compatible with prior studies in settings including macroautophagy[22–24], bacterial[27], fungal[28], and viral infection[27,28] the mechanism of competitive binding of Rubicon to promote formation of new protein complexes appears a generalizable feature for the functionality of this molecule. Competitive binding mediated by Rubicon is spatially constrained to endosome compartments where Rubicon inducibly co-localizes. Endosome localization of Rubicon may dictate its ability to form new complexes based on local protein levels on the signaling endosome, the stoichiometries of which may be dynamically regulated as a result of endosome maturation and/or trafficking. These interactions may involve its CARD domain, shown here to mediate binding to numerous inflammasome components. Thus, endosome localization of Rubicon may be linked with its ability to form new protein complexes, a process critical for its effector function. As Rubicon is a Rab7 effector and is expected to show binding to late endosomes, various studies including our own have shown that Rubicon localizes to early endosomes. Our studies may explain these paradoxical findings as Rubicon binding to ZFYVE21 whose localization is confined to early endosomes may regulate its vesicular targeting to early endosomes.

## Methods

### Cell culture, reagents, and culture treatments

All protocols were approved by the Yale IRB. HUVEC were isolated as healthy, de-identified tissues from the Dept of Obstetrics and Gynecology at Yale New Haven Hospital as previously[2,3,9–12]. HEK293 cells were commercially obtained (ATCC). 'High' panel reactive antibody (PRA) sera were obtained as pooled, de-identified sera from the tissue typing laboratory at Yale New Haven Hospital. 'High' PRA sera were taken from renal transplant candidates showing allo-sensitization of ≥80% and negative testing for numerous infectious agents[10]. Prior to use, PRA sera was supplemented with human complement (Sigma). HUVEC were pooled from 3 human donors and cultured in complete EBM media (Lonza) containing bullet supplements (Lonza). For PRA

treatment, HUVEC were pre-treated with IFN-γ (50 ng/mL, Invitrogen) for 48–72 h prior to placement in gelatin veronal buffer (Sigma) at 25% v/v for the indicated times. Where indicated, HUVEC were pre-treated with PIKIII (5 nM, Cayman Chemical) for 30 min prior to PRA treatment.

For immunostaining, HUVEC were grown on glass coverslips, fixed and permeabilized with ice cold methanol for 15 min, blocked with PBS containing 0.1% Tween and 5% FBS for 1 h at room temperature (PBST). Primary antibodies were then incubated overnight at 4 °C at 1:200 dilution using the following antibodies: Ulex (Vector Labs, #B-1065), P65 (Santa Cruz, #sc-8008), ZFYVE21 (Biorbypt, #orb221973), Rubicon (Cell Signaling, #8465), and RNF34 (ProteinTech, #10629-1-AP). The next morning slides were washed 3 times in PBS and incubated 1 h at room temperature with AlexaFluor anti-mouse Ig 546 nm and AlexaFluor anti-rabbit Ig 647 nm antibodies (Molecular Probes) at 1:1000 dilution. Following staining, slides were washed, air dried, and cover slipped using a DAPI mounting media (ImmunoGold with DAPI, Invitrogen). Proximity ligation assays were performed according to the manufacturer's specifications (Sigma). Immunofluorescence was visualized using a Leica SP8 confocal microscope.

For co-immunoprecipitations, cells were lysed in RIPA buffer without SDS (Cell Signaling) containing protease inhibitor tablets (Roche, 1 tablet per 10 mL RIPA buffer) in 1.5 mL Eppendorf tubes with gentle agitation for 1 h at 4 °C. Following this incubation, lysates were pre-cleared with 15 μL protein A/G agarose beads (Pierce) and incubated with 10 μL of antibody against Rab5 (Santa Cruz #sc,8008), Rab7 (Cell Signaling, #9367 S), FliI (Bethyl, #A301-565A-M) or caspase-1 (Santa Cruz, #sc-392736) antibody at 4 °C overnight. The next day samples were incubated with 20–30 μL protein A/G agarose beads for 4 h at 4 °C prior to centrifugation at 5000 rpm and washing using either RIPA buffer without SDS or TBS containing 3% Tween-20 three times prior to Western blotting as below. For whole cell lysates samples were harvested in RIPA buffer (Sigma) containing protease inhibitor tablets (Roche, 1 tablet per 10 mL RIPA buffer) with gentle agitation for 1 h at 4 °C prior to Western blotting.

Following the above, 4X Laemli's buffer (12 μL) and 1 mM DTT (6 μL) were added to 32.5 μL sample, and this mixture was heated for 95 °C for 13 min. Subsequently, samples were loaded onto pre-cast polyacrylamide gels (Bio-Rad), and proteins were electrophoretically separated and transferred to methanol-activated PVDF membranes at 4 °C for 90 min. Membranes were washed for 15 min three times using Tris-buffered saline containing 0.1% Tween-20 pH 7.4 (TBS-T, AmericanBio), blocked with TBST-1 containing 3% bovine serum albumin (Sigma) for 1 h at room temperature, and incubated with primary antibody at 4 °C overnight. Antibodies used for Western blotting were all used at 1:1000 dilution and included ZFYVE21 (Novus Biologicals, #H00079038-B01P), Rubicon (Cell Signaling, #8465), RNF34 (ProteinTech, #10629-1-AP), NIK (Cell Signaling, #4994), Rab5 (Santa Cruz Biotechnology, #sc-46692), K48-conjugated ubiquitin (Cell Signaling, #8081), caspase-1 (Santa Cruz Biotechnology, #sc-392736), cleaved caspase-1 (Cell Signaling, #4199S), and β-actin (Sigma, #A5316-100 μL).

### Cell-free analyses of ZRR complexes

For protein co-incubation studies, Rubcion-GST (1 μg, Origene, #TP320593) and RNF34-His proteins (1 μg, Novus, #NBP2-23440) were co-incubated at 1:1 ratio at a volume of 50 μL at room temperature overnight with gentle rocking prior to resolution by SDS-PAGE. For far Western blots, ZFYVE21-GFP protein ((1 μg, Origene, #TP303337) was added as a probe to Rubicon-GST, and RNF34-His proteins loaded at 1 μg per lane for 4 °C overnight prior to protein resolution and Western blotting under non-denaturing conditions. For agarose bead pulldowns, ZFYVE21-GFP was incubated with 30 μL of protein A/G agarose beads for overnight at 4 °C. The following day, Rubicon-GST and RNF34-His proteins were added separately or co-incubated for 4 h at a 1:1 ratio at room temperature prior to addition to ZFYVE21-GFP-conjugated agarose beads and incubation overnight at 4 °C. All steps

were carried out TBS containing 3% Tween-20. Subsequently, beads were washed with TBS-3% Tween-20 buffer, proteins were eluted using 0.1 M glycine pH3 for 15 min at room temperature with gentle agitation. Proteins were then resolved under non-denaturing conditions and analyzed by Western blot as indicated. For ELISAs, varying concentrations of plate-bound protein as bait were incubated on 96-well flat-bottomed Nunc MaxiSorp ELISA plates in $Na_2CO_3$ pH 9.6 buffer for 4 °C overnight at a final volume of 25 μL with gentle rocking. The following day, plates were washed three times with PBS containing 0.05% Tween-20 (PBS-T), blocked with 5% BSA PBS-T for 1 h at room temperature, and soluble protein as prey was added at varying concentrations in 5% BSA PBS-T at 4 °C overnight at a final volume of 25 μL with gentle rocking. The following day, plates were washed six times with PBS-T and antibody directed against the soluble probe protein was added at 1:1000 concentration and incubated at 4 °C overnight at a final volume of 50 μL. The next day, plates were washed six times and secondary HRP-conjugated antibody was added at 1:5000 at a final volume of 100 μL. Plates were washed 8 times with PBS-T and TMB substrate (R&D Systems) was added at a final volume of 50 μL. Color development of stopped by addition of 50 μL of 2 N $H_2SO_4$, and absorbance was measured at 450 nm.

### siRNA transfection of EC
HUVEC were pre-treated with IFN-γ for 48 h prior to siRNA transfection. siRNA targeting ZFYVE21, Rubicon, and RNF34 or non-targeting siRNA (target sequence UAA CGA CGC GAC GUA A) were purchased as pooled siRNA (Horizon Discovery) and transfected into HUVEC at ~50–70% confluency in 24-well plates (BD Falcon) at 20nM-40nM concentration. siRNAs were diluted at 20 nM concentration in Opti-Mem culture media (Gibco) and mixed at equal volume with RNAiMax transfection reagent (Invitrogen) diluted 1:50 in Opti-Mem for 45 min at room temperature as per the manufacturer's specifications. This mixture was then added to HUVEC cultures at 1:6 ratio for 37 °C for 5–6 h prior to washing and buffer exchange with EGM2. IFN-γ was added at 50 ng/mL, and cells were then analyzed by Western blot, luciferase assay, RT-PCR, or T cell functional assays 48 h later (72 h after transfection).

### Real time quantitative reverse transcription-polymerase chain reaction (quantitative RT- PCR
RNA was isolated from treated HUVEC according to the manufacturer's specifications (Qiagen) and reverse transcribed (Applied Biosystems, Foster City, CA). Respective cDNA was amplified in a CFX Realtime System (Biorad, Hercules, CA) at a volume of 20 μL containing dilutions of 1:20 Taqman probe (Applied Biosystems), 1:2 Taqman Gene Expression Master Mix (Applied Biosystems), and 1:10 cDNA in ddH-2O. RT-PCR gene probes were purchased from Applied Biosystems and included: CCL5 (#Hs00174575_m1), CCL20 (#Hs01011368_m1), IL6 (#Hs00985639_m1), SELE (#Hs00950401_m1), VCAM-1 (#Hs01003372_m1), and GAPDH (Hs02758991_g1). For amplification, samples were heated to 50 °C for 2 min for once cycle, 95 °C for 10 min for one cycle, and then 40 cycles where samples were heated to 95 °C for 15 s proceeded by 60 °C for 1 min.

### FACS-assisted vesicular membrane sorting and analysis
For Rab5-associated speck sorting, HUVECs were stably co-transduced with Rab5-RFP and ASC-GFP and grown in 20 T175 flasks per group. Following PRA treatment for 30 min, HUVECs were harvested, washed, and resuspended in 0.5 mL endosome buffer [10 mM HEPES-NaOH pH 7.4, 1 mM EDTA, 0.25 M sucrose containing 1 protease inhibitor tablet (Roche) per 10 mL buffer] and mechanically disrupted by three freeze-thaw cycles followed by three cycles of sonication for 20 s each, and lysates were ultracentrifuged. GFP+FSC$^{hi}$ events were gated (Supplementary Fig 1b) and isolated by FACS sorting (FACSAria, Becton Dickinson). FACS-sorted vesicles were then subjected to trypsin digestion, and analysis via tandem LC-MS/MS. Tandem mass spectra were extracted and analyzed using Mascot software (Matrix Science).

### Viral transduction of EC
Lentivirus-encoded control shRNA, ZFYVE21 shRNA, Rubicon shRNA, and RNF34 shRNA (Mission, Sigma) were transduced using two cycles of transduction at an MOI of 10 for 8 h each in HUVEC cultures containing EGM2 medium. Knockdown efficiencies were confirmed by Western blot prior to use in collagen gels in vivo. NF-κB luciferase lentiviral particles were commercially obtained (Cignal Reporter Assay, Qiagen) and used at an MOI of 20 to infect HUVECs for 8 h two times[10]. The following reporter constructs were obtained from Addgene: mCherry-Rab5 WT (a gift from Gia Voeltz, plasmid #49201). mCherry-Rab5 DN (a gift from Sergio Grinstein, plasmid #35139), Rab7 WT-GFP (a gift from Richard Pagano, plasmid #12605), Rab7 DN-GFP (a gift from Richard Pagano, plasmid#12660), Ubiquitin-HA WT (a gift from Rachel Klevit, Addgene plasmid #12647).

### Cyclic immunofluorescence
Cyclic immunofluorescence was performed for analysis of tissue biopsies. Human biopsy samples were embedded in FFPE and sequentially hydrated. Samples were blocked in blocking buffer (0.1% PBS-Tween-20 containing 5% FBS) and sequentially stained using goat anti-C6 (Quidel #A307) at 1:200 dilution, mouse anti-ZFYVE21 (Abnova #H000079038-B-1P), rabbit anti-Rubicon (Cell Signaling #8465S) used at 1:100, an antibody cocktail containing CD31 (R&D #FAB3628V) at 1:500 dilution and DAPI (Thermofisher, #62248) used at 1:1000. The following day, sections were washed and secondary antibodies were added in blocking buffer at 1 hr at room temperature. Secondary antibodies were added at 1:1000 dilution and included anti-mouse AF488 Ab (ThermoFisher # A-21202), anti-rabbit AF555 (ThermoFisher # A-31572), and anti-goat AF647 Ab (ThermoFisher #A-21447). After tissue staining, coverslips were placed on slides containing 10% glycerol in PBS, and imaged (Leica SP8). Subsequently, coverslips were removed by gentle agitation in PBS for 15 min, and sections were bleached with 3% $H_2O_2$ and 20 mM NaOH in PBS for 1 hr while exposed to light for fluorescence quenching. Sections were then restained with primary and secondary antibodies. Images were manually aligned using ImageJ. This multiparameter analysis enabled us to simultaneously visualize colocalization of ZRR complexes with complement-bound capillary beds.

### EC:T cell cocultures
All protocols were approved by the Yale Institutional Review Board (#0601000969). PBMCs were isolated from leukopacks using density centrifugation as described previously and cryopreserved in liquid nitrogen[10]. CD4+CD45RO+T cells were isolated from thawed cryovials using magnetic bead separation kits (Miltenyi) with HLA-DR Ab (clone L243, Novus #NB100-77855) and CD45RA Ab negative depletion (10 μL per cryovial, eBiosciences, 14-0458-82). For EC:T cell cocultures, HUVEC isolated from a single donor were grown in U-bottom 96-well microtiter plates, pretreated with human IFN-γ (50 ng/mL, Invitrogen) for 48–72 h. On the day of the experiment, ECs were placed in gelatin veronal buffer containing 25% v/v of PRA sera for 6 h prior to addition of human CD4+CD45RO+T cells which were added at 0.5–1 × 106 cells/well at a volume of 200 μL in RPMI (Gibco) supplemented with 5% FBS, 1.5% L-glutamine, and 1% penicillin/ streptomycin. T cells were harvested 10–14 days after co-culture in a humidified incubator at 5% CO2 and 37 °C prior to FACS analysis.

### Collagen-fibronectin gel in vivo studies
All protocols were approved by the Yale IACUC (#2021-20175). HUVEC were grown to confluency in 1 T75 flask per mouse. HUVECs were harvested, pelleted, and resuspended in 500 μL of a solution mixture containing 50 μL 10X M199 (Sigma), 50 μL of 1 mg/mL

fibronectin (Millipore), 12.5 µL 1 M HEPES, 188 µL of 10 mg/mL NaHCO₃, 300 µL of 3.66 mg/mL Collagen (Corning), and 6 µL 1 M NaOH. Gels were solidified at 37 °C for 2–3 h, labeled with India ink to aid visualization upon harvesting, and subcutaneously implanted into the flanks of SCID/bg mice. Four weeks later, gels were harvested, flash frozen in OCT, sectioned, stained, and analyzed by I.F. as indicated. Staining Abs were used at 1:200 dilution including ZFYVE21 (Atlas, #055721), Rubicon (Cell Signaling, #8465), and RNF34 (ProteinTech, #10629-1-AP), cleaved caspase-1 (Cell Signaling, #4199S), and Ulex (Vector Labs).

## Murine in vivo studies

All protocols were approved by the Yale IACUC (#2021-20175). Adult 6–12 week old C57/Bl6 and Rubicon-/- mice (Jackson Labs) were injected i.v. via tail vein injection with 750 µg anti-mouse MOPC Ab (Ichor, clone MPC-11) or 750 µg anti-H-2$^b$ Ab (Ichor, clone 8-24-3). Twenty-four hours later, kidneys were harvested for I.F. analyses and co-IP studies. For I.F. analyses, skin grafts were flash frozen in OCT, sectioned and processed as above, and stained for CD31 (Novus, #AF3628) and C6 (ProteinTech, #17239-1-AP).

In separate experiments, skin from male C57/Bl6 or Rubicon-/- (H-2$^b$) mice treated with MOPC Ab or anti-H-2$^b$ Ab as above were harvested 24 h after Ab injection and implanted on the dorsal flanks of female SCID/bg hosts. Seven days later, mice were injected i.p. with $1 \times 10^6$ female C57/Bl6 splenocytes. Skin grafts were harvested three weeks later for analysis.

Human coronary arteries were interposed into the infrarenal abdominal aortae of adult 6–12-week-old female C.B-17 SCID/beige mice (Taconic, #CBSCBG-F) for ~30 days to allow for tissue quiescence. Mice were then given i.v. injection of 200 µL neat PRA sera or IgG-depleted PRA sera as controls, and grafts were harvested 24 h later for proximity ligation analyses as per the manufacturer's specifications (Sigma).

## Statistical methods

Paired analyses were performed using two-tailed Student's *t* test and multiple comparisons were performed using a one-way or two-way ANOVA followed by Tukey's pairwise comparison test using Origin computer software. *p*-values < 0.05 were considered statistically significant. Standard deviations are reported throughout the text.

## Reporting summary

Further information on research design is available in the Nature Portfolio Reporting Summary linked to this article.

## Data availability

The following transcriptomic datasets were retrieved from the Gene Expression Omnibus: GSE147089 (*n* = 224), GSE112943 (*n* = 21), and GSE777298 (*n* = 23). All data are available from the authors upon reasonable request. Source data are provided with this paper.

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

## Acknowledgements

D.J. was supported by grants from the NIH (R01HL141137), Veteran's Administration (I01BX005117), and the American Lung Association (ETRA 736563). J.S.P. received funding from the NIH (U01AI132895). M.F. was supported by the First-Year Research Fellowship, an internal grant from Yale College. All authors declare no financial conflicts of interest.

## Author contributions

X.L., Q.J., S.G., M.N.B., and C.F. performed Western blot experiments. Q.J. performed truncation mutant studies and molecular cloning. B.J., A.G., and G.T. were involved in skin retransplantation and human artery retransplantation studies. S.W. and J.J. were involved in human biopsy analysis. M.F. performed public database analyses. A.G. and G.T. provided human tissues. J.P. assisted in procuring biopsy specimens. Q.W. performed manuscript edits. D.J. wrote the manuscript, performed Western blot experiments, I.F., and qRT-PCR studies, drafted the manuscript, designed all experiments. X.L. and D.J. analyzed all experimental data.

## Competing interests

The authors declare no competing interests.
