## [Peer Review File · Nature Communications]

REVIEWER COMMENTS

Reviewer #1 (Remarks to the Author):

The manuscript by Li et al describes the assembly of a ZFYVE21-Rubicon-RNF34 signaling complex in endothelial cell inflammasome activity produced by membrane attack complexes (MACs). This group has been advancing the understanding of inflammasome formation and activity in endothelial cells that play a role in systemic inflammation and acute rejection. Building on previous work that has shown NLRP3, and CARD domain proteins promote protein aggregation in inflammasome activity, resulting in the processing of pro-IL1b and IL18.

In this study, the authors extend their previous analysis of the proteomics of isolated inflammasomes in endothelial cells treated with antibodies from acute rejection. They describe the binding of a Rab5 effector, ZFYVE21, Rubicon and RNF34 on early endosomes. The complex dissociates Caspase 1 from a pseudosubstrate, flightless (Fli), and promotes its degradation and caspase cleavage. Data from models of antibody mediated rejection are provided that support the role of this complex in inflammasome action in vivo.

This is an insightful manuscript that incorporates informatics sleuthing and a number of cell biology and protein interaction studies to advance the mechanism for how ZFYVE21 plays a role in caspase 1 activation/inflammasome activation. The strengths include the significance of the topic, and the multiple approaches (PLA, colIP, colocalization and far-western blot) to demonstrate the formation of the heterotrimeric ZFYVE21-Rubicon and RNF34 complex (ZRR). The effect of the Rab5DN mutant on ZFYVE21-Rubicon and RNF34 complex recruitment is dramatic, as is the loss of Fli in rab5 co-IPs. The extensive demonstration that this complex forms in a number of in vivo systems including antibody mediated rejection SLE and chronic rejection demonstrates clinical and translational relevance.

Major comments

In Figure 1b, is the data of sufficient resolution to conclude whether NIK activation occurs after formation of Caspase 1?

Much of the complex formation data uses IFN priming to activate MHC expression, as well as the protein components of the inflammasome. Is the role of ZFYVE21-Rubicon-RNF34 complex only seen in the presence of IFN priming in vivo? Are IFNs activated in the human/animal model? This might be worth discussion.

The conclusions from Figure 2e, where the effects of CHX on reducing Rubicon and RNF34 are interpreted to be from post-translational mechanisms is confusing to me. This result is also compatible with inhibiting a transcriptional induction of these proteins.

The loss of Fli in rab5 colPs is robust, but some of the data in Fig 4 is less convincing. For example, the IF images in Fig. 4C showing interaction with Fli and rab5 vesicles by IF are difficult to interpret. What is the magnification of these images, and what are these globular structures? The induction of CCL20 and SEL cytokine secretion is unconvincing in my view, with influence from a few outlying points. Why did the authors not measure the secretion of IL1B or IL18?

Overall, this is a complicated and densely written paper with multiple supporting citations that may make it difficult for the general reader of the journal to access. Consequently, the major unresolved question addressed in the paper about the mechanism of ZFYVE21 action in inflammasome activation is not strongly stated. Some comments that the authors might consider:

The introduction, for example, has an extensive half-page description of complement activation in MAC formation, and extensive review of this groups prior work. This description undermines the impact/novelty of the current study, making this substantial work appear somewhat incremental.

The comments on line 155 about searching for FYVE/FYVE-like domains is not clear until the rationale is described later on line 246.

The abbreviation CABMR is introduced on line 371 without definition in the text or legend of figure 6.

The focus of this paper is entirely on MAC complexes, but a paragraph of the very short discussion talks about responses to candida infections. This seems somewhat peripheral, without more information on whether MAC complexes play a role in response to candida.

Reviewer #2 (Remarks to the Author):

In the present manuscript, the authors describe the discovery of a ternary complex (termed ZRR complex) between ZFYVE21, the RAB effector protein Rubicon and the Ubiquitin ligase RNF34. This complex is recruited to endosomes in a RAB5 dependent manner upon treatment of endothelial cells with panel reactive antibodies (PRA) from patient sera. Upon induction with PRA the ZRR complex

displaces the Caspase 1 inhibitor Flightless/Fli1 from Caspase 1 through direct binding to Fli1 and also through ubiquitination of FLI1. This results in enhanced Caspase 1 cleavage and subsequent production of inflammatory cytokines such as IL1beta and IL18. They proceed to demonstrate that this complex formation also plays a role in inflammation in several in vivo mouse models, which very nicely corroborates their in vitro findings.

Overall, the data appears to be of high quality. The experiments follow a logical flow and the conclusions derived from the data look sound to this reviewer. The merging of primary biochemical data from cultured cells and the subsequent confirmation in living mice is an impressive array of experimentation that clearly demonstrates a vital role for this newly identified complex in several inflammatory diseases. I do have some relatively minor points that the authors should address but I am otherwise very supportive of publication.

Individual points:

Figure 2 and 3: The authors should corroborate their biochemical experiments regarding rubicon and RNF34 recruitment with imaging based approaches. Do ZFYVE21, Rubicon or RNF34 lose their endosomal association with DN-Rab5 (or Rab5 knockdown?). Similarly, does knockdown of Rubicon cause the microscopic loss of RNF34 from RAB5 positive endosomes?

Figure 1C: Given that PLA assays often generate false positives, the authors should also use a negative control. They could test whether another RAB such as RAB11 also generates a PLA signal with Rubicon.

Figure 1D: The authors should show a magnification of an inset to better visualize the colocalization on an endosome.

Figure 1 and 2: Why do the authors jump from describing data shown in Figure 2 back to data shown in Figure 1? This is highly confusing. The authors should rewrite the text and or rearrange the figures to address the data in the figures in a chronologically more coherent way.

Figure 3h: Is the visible band with the AA252-364 bait construct a background level of binding? The authors should repeat this experiment and include a proper control such as a GFP only. The same

applies to Figure 3g, a GFP only control would indicate whether there is background binding to the beads.

Reviewer #3 (Remarks to the Author):

Comments to Editors:

The manuscript entitled “A ZFYVE21-Rubicon-RNF34 Signaling Complex Promotes Endosome-Associated Inflammasome Activity in Endothelial Cells” describes a mechanism in which the internalization of membrane attack complexes (MAC) stimulates inflammasome activation through an endosomal associated process that includes ZFYVE21, Rubicon, and RNF34. The data presented in the manuscript is sufficient to demonstrate how this tripartite ZRR complex is stabilized on Rab5+MAC+ vesicles and facilitates RNF34-mediated ubiquitination and proteasomal degradation of the endogenous caspase-1 inhibitor Flightless I. Active caspase-1 levels are consequently elevated, supporting previous work that established the secretion of inflammasome associated cytokines, IL-1 β and IL-18 in endothelial cells in response to MAC internalization. Extensive data from various experiments support the author’s hypothesis. Minor to moderate concerns were noted and are listed below.

Major comments:

- 1) Under the inflammatory condition, are there any Rab5+caspase-1+ endosomes without MAC, and vice versa?
- 2) Since the Rubicon function is associated with autophagy inhibition, are these Rubicon functions described in the manuscript dependent or independent of autophagy?
- 3) Can authors show if the NLRP3/ASC inflammasome complex interacts with the ZRR complex on Rab5+MAC+ endosomes to activate caspase-1?
- 4) On the same note, it is important to determine if early endosome associated caspase-1 is also bound to NLRP3/ASC or other inflammasomes?
- 5) Rab5 activity is preserved in Rab5+MAC+ vesicles (Fig. 2a-b). Do Rab5+ early endosomes maintain their pH after fusion with MACs endocytosed vesicles? pH would be an indicator of endosome damage.
- 6) Fig. 4b: CARD16 has multiple non-specific bands in the original western blot in Fig. S11. Please address this concern.
- 7) Can authors show colocalization of Rab5 with ZFYVE21, Rubicon, and RNF34 in Fig. 6d-f and Fig. 7a?

Minor comments:

- 1) Label the western blot images with appropriate molecular weights in Fig. 3b-c. Clarify why there is no Rubicon-GFP band above 100 kDa in the original western blot image shown in Fig. S10 meant to support Figure 3b. Please also clarify if GAPDH is input loading control or IP in Fig. 3c. Figure 3e (50 kDa ZFYVE21-GFP) does not appear to be the original western blot data in Fig. S10. Please correct in Fig. S10. Double-check other blots for mismatched labeling.
- 2) Fig. 3e, and page 11 and lines 24-26: Clarify how the Rubicon and RNF34 heterodimer (150 kDa) is stabilized in the denaturing western blot shown in Fig. 3e. How was the coimmunoprecipitated Rubicon and RNF34 heterodimer eluted from ZFYVE21-GFP immobilized agarose beads? Can authors confirm the presence of Rubicon and RNF34 in a heterodimer complex with mass spectrometry?
- 3) Fig. 3f: Specify the absorbance units in the graph and add a subsection detailing this experiment in the Methods section.
- 4) Fig. 3g-h: Specify the name of the protein that corresponds to the domain organization in Fig. 3h. Do this for any similar figures. Instead of Input: GFP, please change the labels to Rubicon-GFP and RNF34-GFP, respectively.
- 5) Page 13 and line 3: Please clarify what lanes are being compared in Fig. 4e (lane 1 vs. lane 3 or lane 1 vs. lane 4)?
- 6) Page 14 and lines 12, 14: This should read E2 conjugating enzymes instead of E2 ubiquitin ligases.
- 7) Page 14, 15: The subtitles "ZRR Complexes Form in Patient Tissues, and ZRR complexes form in human tissues" are misplaced. Please correct.
- 8) Page 18 and line 19-20: It should be (Fig 8h)
- 9) Discussion, page 18 and line 12: It should be (Fig 8h)
- 10) Fig. 7h: Input labeling is misplaced. It should be IB (Lysate?) and Input, respectively.
- 11) Add the appropriate scale bar in all images that contain microscopy.
- 12) Label the western blot images in the main figures with the molecular weight of the bands to assist the reader.

REVIEWER COMMENTS

To address reviewer concerns, we have re-written the Introduction and Discussion sections in their entirety, and we have made extensive edits to the Results, Methods, and Discussion sections. All of the panels of the Main Figures 1-8 have been modified from the prior version of this manuscript to improve clarity of presentation. Additionally, we have added 7 new experiments to the Main Figures, and we have added 15 new Supplementary Figures excluding the statistical quantifications and uncropped Western blots. We hope that these extensive changes have addressed the reviewers' concerns. We discuss point-by-point the reviewers' concerns below.

Reviewer #1 (Remarks to the Author):

The manuscript by Li et al describes the assembly of a ZFYVE21-Rubicon-RNF34 signaling complex in endothelial cell inflammasome activity produced by membrane attack complexes (MACs). This group has been advancing the understanding of inflammasome formation and activity in endothelial cells that play a role in systemic inflammation and acute rejection. Building on previous work that has shown NLRP3, and CARD domain proteins promote protein aggregation in inflammasome activity, resulting in the processing of pro-IL1b and IL18.

In this study, the authors extend their previous analysis of the proteomics of isolated inflammasomes in endothelial cells treated with antibodies from acute rejection. They describe the binding of a Rab5 effector, ZFYVE21, Rubicon and RNF34 on early endosomes. The complex dissociates Caspase 1 from a pseudosubstrate, flightless (Fli), and promotes its degradation and caspase cleavage. Data from models of antibody mediated rejection are provided that support the role of this complex in inflammasome action *in vivo*.

This is an insightful manuscript that incorporates informatics sleuthing and a number of cell biology and protein interaction studies to advance the mechanism for how ZFYVE21 plays a role in caspase 1 activation/inflammasome activation. The strengths include the significance of the topic, and the multiple approaches (PLA, coIP, colocalization and far-western blot) to demonstrate the formation of the heterotrimeric ZFYVE21-Rubicon and RNF34 complex (ZRR). The effect of the Rab5DN mutant on ZFYVE21-Rubicon and RNF34 complex recruitment is dramatic, as is the loss of Fli in rab5 co-IPs. The extensive demonstration that this complex forms in a number of *in vivo* systems including antibody mediated rejection SLE and chronic rejection demonstrates clinical and translational relevance.

We thank the reviewer for the remarks above and for the suggestions as addressed point-by-point below. For clarification purposes, we have also included a description of 'high' PRA sera which was added to the Introduction:

Although MACs may lyse microbes or red blood cells, nucleated cells like ECs typically resist MAC-induced lysis by various mechanisms including both shedding of membrane vesicles and by internalization.¹³ We used 'high' panel reactive antibody (PRA) sera obtained from renal transplant candidates to model the non-cytolytic properties of MACs on human ECs *in vitro* and *in vivo*^{7-8,14-17}. Due to risks of MACs, transplant candidates routinely undergo PRA testing where patient sera is overlaid on a broad panel of Luminex beads, each of which is coated with a singular HLA specificity. The number of beads showing reactivity is reported as a percentage reflective of prospective risk of developing alloAb-induced MACs on ECs upon transplantation. Due to multiple childbirths, blood transfusions, and/or prior transplants, certain patients contain high titers of non-self binding alloantibodies (alloAbs) showing >80% bead binding. These patients with 'high' PRA sera are especially at-risk for developing MAC formation on ECs to cause tissue rejection. We used 'high' PRA sera, termed PRA, to elicit alloAb-induced and non-cytolytic assembly of MACs on ECs.

Major comments

In Figure 1b, is the data of sufficient resolution to conclude whether NIK activation occurs after formation of Caspase 1? This is a good question, and we have attempted to address this in prior reports. In ref 6,7, and 14 of the manuscript, we showed that ZFYVE21 is required for NIK to colocalize with signaling endosomes, and that NIK colocalization on signaling endosomes is required for caspase-1 activity. To confirm these findings, we performed siRNA-mediated knockdowns of NIK and reciprocal siRNA knockdowns of caspase-1 as shown in Supplementary Fig 5b,c and whose statistical quantifications are shown in Supplementary Fig 4f,g. We observed that loss of either NIK or caspase-1 decreased cleaved caspase-1, but showed no effects on the stability of ZRR complex proteins.

Much of the complex formation data uses IFN priming to activate MHC expression, as well as the protein components of the inflammasome. Is the role of ZFYVE21-Rubicon-RNF34 complex only seen in the presence of IFN priming *in vivo*? Are IFNs activated in the human/animal model? This might be worth discussion.

This is an interesting point to consider. In addition to the plethora of signals including PAMPs, DAMPs, hypoxia, and metabolites which may prime NLRP3 inflammasomes, we and others have observed that IFN- γ primes NLRP3 inflammasomes during the sterile inflammatory response elicited with transplantation rejection. PRA-treated ECs show enhanced immunogenicity and elicit strong EC-mediated type 1 responses in alloimmune CD4⁺ T cells. This has been demonstrated in EC:T cell coculture experiments *in vitro* and in humanized mouse models *in vivo* where we have reproducibly observed potentiated IFN- γ responses by alloimmune CD4⁺ T cells as a result of PRA treatment of target ECs.

The ZRR complex is newly discovered and as of yet we do not yet know its full role with relation to priming by IFN- γ and/or other self/non-self molecules. While we have found that IFN- γ priming is required for the functional role of ZRR complexes, our studies do not rule out contribution(s) from other endogenous molecule eliciting inflammasome priming. We have included the following text in the Discussion on page 20:

An antecedent priming step upregulates and licenses inflammasome proteins for assembly. While induction of inflammasome proteins canonically occurs following PAMP-induced NF- κ B, this process may be subsumed by endogenous molecules including reactive oxygen species, metabolites, and hypoxia during transplant-associated sterile inflammation.⁴⁸ Various cytokines including IFN- γ , though dispensable for IL-1 β production in human macrophages, may similarly prime NLRP3 inflammasomes,⁴⁹ a process that we found occurs following EC-mediated direct allorecognition *in vitro* and in various humanized mouse models including coronary artery xenografts^{7,14,50} *in vivo*. While abundant IFN- γ is induced in our *in vitro* and *in vivo* approaches above, our studies do not rule out the contribution(s) from other endogenous molecules eliciting inflammasome priming. Our data herein using murine and humanized models show that *in situ* levels of endogenous priming molecule(s) are sufficient for generating IL-1 β by ECs *in vivo* in the absence of an experimentally-induced priming step. As the ZRR complex is newly discovered, defining such signals with regards to the function of this complex will be required to fully define the scope of its function(s) *in vivo* and is the focus of ongoing studies by our group.

The conclusions from Figure 2e (now Figure 2h), where the effects of CHX on reducing Rubicon and RNF34 are interpreted to be from post-translational mechanisms is confusing to me. This result is also compatible with inhibiting a transcriptional induction of these proteins. In this experiment, PRA was added at the times indicated in the presence or absence of cycloheximide, a protein synthesis inhibitor. Without cycloheximide, ZRR complex-associated proteins became upregulated within 30 min, a kinetic timeframe compatible with post-translational protein stabilization. The stability of these proteins at timepoints >30 min were reduced by CHX, indicating that ongoing protein synthesis was required to sustain levels of ZRR complex-associated proteins. NIK, a protein whose levels are known to be post-translationally regulated, is included as a control showing similar effects, i.e., decreased stability with CHX treatment. These findings are supported by Fig 2i showing that treatment with MG132, a proteasome inhibitor, rapidly increased levels of ZRR complexes within

30 min, a timepoint where inflammasome activity was observed.

The experiment in Fig 2h was performed 3 times, and after reviewing the original films shown above, we retain the belief that our results are compatible with our original interpretation that ZRR complexes are post-translational stabilized. If the reviewer wishes, we may remove Figure 2h as Figure 2i also supports the conclusion that levels of ZRR complexes are regulated via post-translational stabilization.

The loss of Fli in rab5 colPs is robust, but some of the data in Fig 4 is less convincing. For example, the IF images in Fig. 4C showing interaction with Fli and rab5 vesicles by IF are difficult to interpret. What is the magnification of these images, and what are these globular structures? The induction of CCL20 and SEL cytokine secretion is unconvincing in my view, with influence from a few outlying points. Why did the authors not measure the secretion of IL1B or IL18? We have repeated these studies using Fli1 siRNA and assessing supernatant levels of IL-1 β using Western blot densitometries as shown in Fig 4g and whose quantifications are shown in Supplementary Fig 8e. We found that loss of Fli1 significantly increased IL-1 β levels, compatible with prior reports showing that Fli1 endogenously inhibits caspase-1 activity.

Overall, this is a complicated and densely written paper with multiple supporting citations that may make it difficult for the general reader of the journal to access. Consequently, the major unresolved question addressed in the paper about the mechanism of ZFYVE21 action in inflammasome activation is not strongly stated. Some comments that the authors might consider: We thank the reviewer for this suggestion, and the Introduction has been rewritten in its entirety on pages 4-5 as highlighted in red in an attempt to make the manuscript more accessible to the general reader. As per suggestions by Reviewer #3, we have also made various labeling changes in an effort to make the manuscript more accessible.

The introduction, for example, has an extensive half-page description of complement activation in MAC formation, and extensive review of this groups prior work. This description undermines the impact/novelty of the current study, making this substantial work appear somewhat incremental. To heighten impact of our findings, in the Introduction we have shortened the description of our prior work. The overall length of the Introduction is slightly increased due to the inclusion of a paragraph describing 'high' PRA sera as above which was included to improve clarity.

The comments on line 155 about searching for FYVE/FYVE-like domains is not clear until the rationale is described later on line 246. For clarification, we have included a new paragraph description highlighted in red on page 6.

The abbreviation CABMR is introduced on line 371 without definition in the text or legend of figure 6. We included this abbreviation on page 13.

The focus of this paper is entirely on MAC complexes, but a paragraph of the very short discussion talks about responses to candida infections. This seems somewhat peripheral, without more information on whether MAC complexes play a role in response to candida. We have removed the peripheral discussion of candida infection from the Discussion. In its place, we describe how our work informs future directions for understanding how endosome contribute to inflammasome activity and how priming responses *in vivo* may affect role(s) of ZRR complexes as per the suggestion above.

Reviewer #2 (Remarks to the Author):

In the present manuscript, the authors describe the discovery of a ternary complex (termed ZRR complex) between ZFYVE21, the RAB effector protein Rubicon and the Ubiquitin ligase RNF34. This complex is recruited to endosomes in a RAB5 dependent manner upon treatment of endothelial cells with panel reactive antibodies (PRA) from patient sera. Upon induction with PRA the ZRR complex displaces the Caspase 1 inhibitor Flightless/Fli1 from Caspase 1 through direct binding to Fli1 and also through ubiquitination of FLI1. This results in enhanced Caspase 1 cleavage and subsequent production of inflammatory cytokines such as IL1beta and IL18. They proceed to demonstrate that this complex formation also plays a role in inflammation in several in vivo mouse models, which very nicely corroborates their in vitro findings.

Overall, the data appears to be of high quality. The experiments follow a logical flow and the conclusions derived from the data look sound to this reviewer. The merging of primary biochemical data from cultured cells and the subsequent confirmation in living mice is an impressive array of experimentation that clearly demonstrates a vital role for this newly identified complex in several inflammatory diseases. I do have some relatively minor points that the authors should address but I am otherwise very supportive of publication.

We thank the reviewer for these encouraging comments, and we have performed experiments to address the reviewer's concerns below.

Individual points:

Figure 2 and 3: The authors should corroborate their biochemical experiments regarding rubicon and RNF34 recruitment with imaging based approaches. Do ZFYVE21, Rubicon or RNF34 lose their endosomal association with DN-Rab5 (or Rab5 knockdown?). Similarly, does knockdown of Rubicon cause the microscopic loss of RNF34 from RAB5 positive endosomes? **These new data have been added in Fig 2a. Based on the reviewer's suggestion, we have also incorporated similar confocal I.F. readouts in Fig 1f, 1i, and Fig 2e. Of note, prior polyclonal Abs which specifically labeled Rubicon and RNF34 in the past, no longer show specific staining, possibly as a result of company changes to Ab production following the COVID-19 lockdown. As such, we have used overexpression studies in the new figures above.**

Figure 1C: Given that PLA assays often generate false positives, the authors should also use a negative control. They could test whether another RAB such as RAB11 also generates a PLA signal with Rubicon. **Rab11 marks a subset of recycling endosomes which contain Rab5, and in pilot studies we observed proximity-dependent MFIs for Rab5 and Rab11. As a negative control, we thus repeated the PLA assays using a nuclear antigen, Histone H3, and this negative control was used in PLAs in Fig 1c and in Fig 4d.**

Figure 1D: The authors should show a magnification of an inset to better visualize the colocalization on an endosome. **New insets are shown in Fig 1d.**

Figure 1 and 2: Why do the authors jump from describing data shown in Figure 2 back to data shown in Figure 1? This is highly confusing. The authors should rewrite the text and or rearrange the figures to address the data in the figures in a chronologically more coherent way. **The relevant text has been moved so that the figures occur in the chronological order in which they are described in the Results.**

Figure 3h: Is the visible band with the AA252-364 bait construct a background level of binding? The authors should repeat this experiment and include a proper control such as a GFP only. The same applies to Figure 3i, a GFP only control would indicate whether there is background binding to the beads. **We thank the reviewer as the prior lack of the GFP only control may have confounded our results. The relevant experiments in Fig 3h and Fig 3i incorporate GFP controls, and the results show increased binding above that observed with GFP only controls.**

Reviewer #3 (Remarks to the Author):

The manuscript entitled "A ZFYVE21-Rubicon-RNF34 Signaling Complex Promotes Endosome-Associated Inflammasome Activity in Endothelial Cells" describes a mechanism in which the internalization of membrane attack complexes (MAC) stimulates inflammasome activation through an endosomal associated process that includes ZFYVE21, Rubicon, and RNF34. The data presented in the manuscript is sufficient to demonstrate how this tripartite ZRR complex is stabilized on Rab5+MAC+ vesicles and facilitates RNF34-mediated ubiquitination and proteasomal degradation of the endogenous caspase-1 inhibitor Flightless I. Active caspase-1 levels are consequently elevated, supporting previous work that established the secretion of inflammasome associated cytokines, IL-1 β and IL-18 in endothelial cells in response to MAC internalization. Extensive data from various experiments support the author's hypothesis. Minor to moderate concerns were noted and are listed below.

We thank the reviewer for the helpful comments and for the careful review of the manuscript. Based on the reviewer's perusal, we have performed experiments as below and have made appropriate corrections to the text and figures.

Major comments:

1) Under the inflammatory condition, are there any Rab5+caspase-1+ endosomes without MAC, and vice versa? We stably co-transduced ECs with Rab5 WT and caspase-1 RFP and stained ECs with C9, the most abundant component of MACs. In these studies we observed both Rab5+caspase-1+ vesicles both containing and lacking MAC as shown in Supplementary Fig 5g.

2) Since the Rubicon function is associated with autophagy inhibition, are these Rubicon functions described in the manuscript dependent or independent of autophagy? To address this question, we first asked whether autophagy was induced by PRA. In time course experiments, we found that p62 and LC3-II became upregulated over time with PRA.

We next asked whether autophagy was required for formation of ZRR complexes and/or activation of caspase-1. To answer this, we transfected PRA-treated HUVECs with control siRNA or siRNA against ATG16L and p62, proximal and terminal autophagy-associated molecules, respectively. We found that knockdowns involving either ATG16L or P62 showed no effect on the stability of inflammasome proteins, ZFYVE21, Rubicon, RNF34, nor on caspase-1 cleavage. This data indicated that autophagy as a process did not proximally regulate ZRR complexes nor inflammasome activity.

To corroborate the above, we induced non-selective autophagy in HUVECs via serum starvation, a process upregulating LC3-II. Induction of non-selective autophagy showed no effects on levels of ZRR complexes and did not induce caspase-1 cleavage. Rubicon siRNA potentiated LC3-II as previously reported but despite increased levels of autophagy as marked by increased LC3-II, ZFYVE21 and RNF34 levels remained unchanged. Moreover, siRNA-mediated inhibition of P62, while ablating LC3-II, did not show effects on ZRR complexes. These data indicated that ZRR complexes and inflammasome activity were not induced or regulated by non-selective autophagy. Together we induced autophagy via PRA and via starvation, and neither of these processes appeared to affect ZRR complex-induced inflammasome activity.

3) Can authors show if the NLRP3/ASC inflammasome complex interacts with the ZRR complex on Rab5+MAC+ endosomes to activate caspase-1? We newly performed Rab5 co-IPs and in the same PRA-treated lysate we detected Rab5-associated ZRR complexes as well as components of the NLRP3 inflammasome including NLRP3, ASC, and caspase-1 as shown in Supplementary Fig 5f. This indicates that all components of NLRP3 inflammasomes and ZRR complexes become associated with Rab5. We newly showed in Fig 2d that ZFYVE21, Rubicon, and RNF34 triply colocalize on the same vesicles and that ZRR complex proteins pulldown with caspase-1, a component of the inflammasome, in Fig 5b. In ELISAs, we performed new experiments to address whether ZRR complexes may interact with NLRP3 and ASC. Plate-bound Rubicon and ZFYVE21 did not demonstrate binding to NLRP3 or ASC, whereas RNF34 showed binding to ASC only. These new data are shown in Supplementary Fig 5h. These data together support interactions among ZRR complexes and inflammasome proteins.

4) On the same note, it is important to determine if early endosome associated caspase-1 is also bound to

NLRP3/ASC or other inflammasomes? This is an important question that we attempted to address in a prior publication showing lack of NLRC4 inflammasome formation with PRA treatment.⁷ To extend these results, we performed pulldowns of Rab5 and reciprocal pulldowns of caspase-1. We found that NLRP3 inflammasome proteins (NLRP3 and ASC) inducibly form in association with Rab5 and ZRR complexes. Here, we found that caspase-1 associated with Rab5, NLRP3 and ASC, but not NOD1, AIM2, or NLRC4. These data indicate solitary formation of NLRP3 inflammasomes following PRA treatment.

5) Rab5 activity is preserved in Rab5+MAC+ vesicles (Fig. 2a-b). Do Rab5+ early endosomes maintain their pH after fusion with MACs endocytosed vesicles? pH would be an indicator of endosome damage. We corresponded with the editor regarding how to address this question as endosome acidification and corresponding change in pH is a feature intrinsic to maturing endosomes and not necessarily specific to damaged endosomes. We believe the reviewer wishes to test whether Rab5+MAC+vesicles colocalize with markers of endosome damage. To answer this question, we co-stained PRA-treated HUVECs with galectin-3, a marker of damaged endosomes. We found that galectin-3 showed high co-localization with C9, a marker for MAC, indicating that endosomes containing MAC show signs of membranous injury. This new experiment is shown in Supplementary Fig 5a.

6) Fig. 4b: CARD16 has multiple non-specific bands in the original western blot in Fig. S11. Please address this concern. We reprobbed the same lysates using a different CARD16 antibody (Aviva Systems), and we observed that the higher molecular weight bands disappeared as shown in Supplementary Figure 4b. We have incorporated the new CARD16 bands at ~22kD in new Fig 4b.

7) Can authors show colocalization of Rab5 with ZFYVE21, Rubicon, and RNF34 in Fig. 6d-f and Fig. 7a? We corresponded with the editor regarding how to address this question as Rab5 is expressed ubiquitously. In response to the reviewers question and as per the editor's instructions, we have performed confocal I.F. of human kidney tissues and found that ZFYVE21 and Rubicon colocalized with CD31+ ECs expressing Rab5 (Supplementary Fig 5i). We noted that MFIs for both ZFYVE21+CD31+Rab5+ and Rubicon+CD31+Rab5+ staining was increased in patients with ABMR compared to healthy controls.

Minor comments:

1) Label the western blot images with appropriate molecular weights in Fig. 3b-c. Clarify why there is no Rubicon-GFP band above 100 kDa in the original western blot image shown in Fig. S10 meant to support Figure 3b. Please also clarify if GAPDH is input loading control or IP in Fig. 3c. Figure 3e (50 kDa ZFYVE21-GFP) does not appear to be the original western blot data in Fig. S10. Please correct in Fig. S10. Double-check other blots for mismatched labeling. In Fig S10, the Rubicon GFP band size has been relabeled as '150kD.' We have double-checked the original blots per the reviewer's request.

2) Fig. 3e, and page 11 and lines 24-26: Clarify how the Rubicon and RNF34 heterodimer (150 kDa) is stabilized in the denaturing western blot shown in Fig. 3e. How was the coimmunoprecipitated Rubicon and RNF34 heterodimer eluted from ZFYVE21-GFP immobilized agarose beads? Can authors confirm the presence of Rubicon and RNF34 in a heterodimer complex with mass spectrometry? Rubicon-GST, and RNF34-His were co-incubated as indicated for 4hr at room temperature prior to incubation with ZFYVE21-GFP-coated protein A/G agarose beads for overnight at 4°C. These steps were all carried out in a non-reducing buffer containing TBS-3% Tween-20. Subsequently, bound proteins were eluted using an acidic glycine elution buffer, and proteins were resolved under non-denaturing conditions and probed as indicated by Western blot. New text was added to the Methods on page 23 and to the figure legend of Main Figure 4 on page 36 for clarification.

To directly test formation of a Rubicon:RNF34 heterodimer, we co-incubated Rubicon and RNF34 proteins as shown in Fig 3d and observed the appearance of a new ~150kD band. This ~150kD band was excised and analyzed by LC-MS/MS which showed 57% and 92% peptide coverage for Rubicon and RNF34, respectively as shown in Supplementary Figure 5d. These data indicated that Rubicon and RNF34 were capable of forming a heterodimer at the predicted molecular weight of ~150kD.

3) Fig. 3f: Specify the absorbance units in the graph and add a subsection detailing this experiment in the

Methods section. The label has been changed to 'Optical Density (450nm)' and the experiment has been detailed in the Methods section on page 24.

4) Fig. 3g-h (New Fig 3h and Fig 3i): Specify the name of the protein that corresponds to the domain organization in Fig. 3h. Do this for any similar figures. Instead of Input: GFP, please change the labels to Rubicon-GFP and RNF34-GFP, respectively. Labels were added to Fig 3h, Fig 3i, and Fig 5c.

5) Page 13 and line 3: Please clarify what lanes are being compared in Fig. 4e (lane 1 vs. lane 3 or lane 1 vs. lane 4)? The text on page 13 has been changed to improve clarity:

Consistent with the above, PRA treatment in the presence of control siRNA caused loss of FliI on Rab5 vesicles, and this occurred concurrently with the appearance of Rab5-associated Rubicon and cleaved caspase-1 isoforms (lane 1 vs lane 3, Fig 4e). Conversely, overexpression of full-length FliI, whose N- and C-terminal domains interact with caspase-1³, reduced Rab5-associated cleaved caspase-1 (Fig 4f, lane 3 vs lane 4). Functionally, FliI siRNA potentiated the generation of elaborated IL-1 β , confirming its described role as an endogenous inhibitor of inflammasome activity (Fig 4g).

6) Page 14 and lines 12, 14: This should read E2 conjugating enzymes instead of E2 ubiquitin ligases. We have made these changes.

7) Page 14, 15: The subtitles "ZRR Complexes Form in Patient Tissues, and ZRR complexes form in human tissues" are misplaced. Please correct. We have made this change.

8) Page 18 and line 19-20: It should be (Fig 8h). The Discussion has been rewritten, and this prior figure reference has been removed.

9) Discussion, page 18 and line 12: It should be (Fig 8i). We have made this change.

10) Fig. 7h: Input labeling is misplaced. It should be IB (Lysate?) and Input, respectively. The word 'Lysate' was changed to 'IB.' in Fig 7h.

11) Add the appropriate scale bar in all images that contain microscopy. The scale bars for all images have been placed.

12) Label the western blot images in the main figures with the molecular weight of the bands to assist the reader. We have inserted molecular weights for all bands in all figures including the new Supplementary Figures.

REVIEWERS' COMMENTS

Reviewer #1 (Remarks to the Author):

This is a nicely revised manuscript, much improved in its clarity and brevity. The authors have adequately responded to the major comments from the first review.

WRT the rebuttal, there are no strong objections to including Fig. 2h with the cycloheximide treatment, but wonder why actinomycin was not used if the authors wanted to show that protein accumulation was independent of new transcriptional synthesis.

Please label the colors for Fli and Rab5 on Fig. 4c; not sure which is which.

On line 320, do the authors mean to refer to "Supplementary" Fig 5g? Otherwise they are citing figures out of order in the main text.

Reviewer #2 (Remarks to the Author):

The authors have performed the additional experimentation that I requested. The imaging data have been improved and proper controls for the immunoprecipitations have been added. As far as I can tell, the authors have also addressed the concerns of the other two reviewers. I am fully supportive of publication now.

Reviewer #3 (Remarks to the Author):

Authors in the revised manuscript entitled "A ZFYVE21-Rubicon-RNF34 Signaling Complex Promotes Endosome-Associated Inflammasome Activity in Endothelial Cells" addressed our prior concerns

reasonably well. However, there is a minor issue in supplementary Fig. 5f in which caspase-1 is seen to be overlapping with denatured antibody heavy chain from immunoprecipitated Rab5 antibody. Can authors confirm whether a strong Western blot protein band is caspase-1 and not denatured antibody heavy chain? Also, include Rab5 Western blot data obtained for IP and input in supplementary Figure 5e & 5f.

REVIEWER COMMENTS

Reviewer #1 (Remarks to the Author):

This is a nicely revised manuscript, much improved in its clarity and brevity. The authors have adequately responded to the major comments from the first review. WRT the rebuttal, there are no strong objections to including Fig. 2h with the cycloheximide treatment, but wonder why actinomycin was not used if the authors wanted to show that protein accumulation was independent of new transcriptional synthesis.

Thank you for your comments as we also feel that the paper will be more accessible to the general reader. As part of our initial phenotyping, treatments with actinomycin, transcriptional inhibitor, were performed as below and no protein changes were detected with this treatment. We did not include these data as cycloheximide and MG132 showed post-translational protein increases, and we were technically unable to perform nuclear run-on experiments to definitively rule out new transcriptional synthesis as a cause for upregulation of these proteins. Kinetically, the rapid increase in ZRR components would strongly suggest post-translational accumulation and not new transcription.

Please label the colors for Fli and Rab5 on Fig. 4c; not sure which is which.

The colors for Fig 4c have been labeled for FliI and Rab5.

On line 320, do the authors mean to refer to "Supplementary" Fig 5g? Otherwise they are citing figures out of order in the main text. Reviewer #2 (Remarks to the Author)

We thank the reviewer for closely reading the manuscript and catching this error. We have changed the text on line 320 to read 'Supplementary Fig 5g' so that the citing figures are in order.

Reviewer #2 (Remarks to the Author):

The authors have performed the additional experimentation that I requested. The imaging data have been improved and proper controls for the immunoprecipitations have been added. As far as I can tell, the authors have also addressed the concerns of the other two reviewers. I am fully supportive of publication now. Thank you very much for your comments which we feel has improved the manuscript.

Reviewer #3 (Remarks to the Author):

Authors in the revised manuscript entitled “A ZFYVE21-Rubicon-RNF34 Signaling Complex Promotes Endosome-Associated Inflammasome Activity in Endothelial Cells” addressed our prior concerns reasonably well. However, there is a minor issue in supplementary Fig. 5f in which caspase-1 is seen to be overlapping with denatured antibody heavy chain from immunoprecipitated Rab5 antibody. Can authors confirm whether a strong Western blot protein band is caspase-1 and not denatured antibody heavy chain? Also, include Rab5 Western blot data obtained for IP and input in supplementary Figure 5e & 5f.

For Supplementary Fig 5e, the experiment was performed per the reviewer’s request to determine the distribution of caspase-1 on signaling endosomes. To validate that the band at 50kD represented caspase-1, we performed caspase-1 siRNAs in PRA-treated ECs prior to performing caspase-1 pulldowns. We noted that the 50 kD band which could putatively represent denatured Ab heavy chain, disappeared with caspase-1 siRNA. The figure is shown below. The antibody we use for caspase-1 co-IPs is (Santa Cruz, clone D3, #sc-392736). The raw Rab5 Western blot data for supplementary Fig 5e and 5f were previously included in supplementary Fig S17 (third row). To highlight this, new text was added on page 18.